# On the Quina side: A Neanderthal bone industry at Chez-Pinaud site, France

**Malvina Baumann** [1]☺*, **Hugues Plisson** [2]☺, **Serge Maury** [3]☺, **Sylvain Renou** [4]‡,
**Hélène Coqueugniot** [2,5]‡, **Nicolas Vanderesse** [2]‡, **Ksenyia Kolobova** [6]‡, **Svetlana Shnaider** [7]‡,
**Veerle Rots** [1]‡, **Guillaume Guérin** [8]‡, **William Rendu** [7]‡

1 TraceoLab, Liege University, Liege, Belgium, 2 PACEA UMR 5199, CNRS, University of Bordeaux, Pessac, France, 3 Les Eyzies-de-Tayac-Sireuil, France, 4 Hadès, Agence Atlantique, Bordeaux, France, 5 Ecole Pratique des Hautes Etudes-PSL University, Paris, France, 6 Paleolithic Department, Institute of archaeology and Ethnography, Siberian Branch of the Russian Academy of Sciences, Novosibirsk, Russian Federation, 7 ZooSCAN, International Research Laboratory 2013, CNRS, Novosibirsk, Russian Federation, 8 Géosciences Rennes UMR 6118, CNRS, University of Rennes, Rennes, France

☺ These authors contributed equally to this work.
‡ These authors also contributed equally to this work
* malvina.baumann@gmail.com

**Data Availability Statement:** All relevant data are within the paper and its Supporting information files.

## Abstract

Did Neanderthal produce a bone industry? The recent discovery of a large bone tool assemblage at the Neanderthal site of Chagyrskaya (Altai, Siberia, Russia) and the increasing discoveries of isolated finds of bone tools in various Mousterian sites across Eurasia stimulate the debate. Assuming that the isolate finds may be the tip of the iceberg and that the Siberian occurrence did not result from a local adaptation of easternmost Neanderthals, we looked for evidence of a similar industry in the Western side of their spread area. We assessed the bone tool potential of the Quina bone-bed level currently under excavation at chez Pinaud site (Jonzac, Charente-Maritime, France) and found as many bone tools as flint ones: not only the well-known retouchers but also beveled tools, retouched artifacts and a smooth-ended rib. Their diversity opens a window on a range of activities not expected in a butchering site and not documented by the flint tools, all involved in the carcass processing. The re-use of 20% of the bone blanks, which are mainly from large ungulates among faunal remains largely dominated by reindeer, raises the question of blank procurement and management. From the Altai to the Atlantic shore, through a multitude of sites where only a few objects have been reported so far, evidence of a Neanderthal bone industry is emerging which provides new insights on Middle Paleolithic subsistence strategies.

## Introduction

Around 48–45 000 BP, the anatomically modern humans (AMH) arrived in Western Europe, while the last Neanderthals disappeared. This arrival resulted in significant changes in archaeological material cultures that define the transition from the Middle to the Upper Paleolithic. AMH brought with them a diversity of artifacts attesting of new practices. Bone materials became a privileged medium for the manufacture of objects, such as hunting weapons,

**Funding:** Authors who did not received a specific funding: H.P.; S.M.; S.R.; N.V.; S.S.; V.R. Funded studies: - M.B. was supported by the Marie Sklodowska-Curie actions under the European Union's Horizon 2020 research and innovation programme (grant agreement #839528) - K.K. was supported by the Russian Science Fundation (grant agreement # 211800376) and the Russian Foundation for Basic Research (grant agreement 195922007) - H.C. was supported by the Fondation Maison des Sciences de l'Homme- W.R. was supported by the French Ministry of Cultural Heritage - G.G. was supported by the European Research Council (ERC) under the European Union's Horizon 2020 research and innovation programme (ERC Grant agreement #851793).

**Competing interests:** The authors have declared that no competing interests exist.

ornaments or figurines, whose functional and social specialization appears clearly through advanced shaping, achieved by scraping or abrasion. In contrast, their absence in the Middle Paleolithic has led to the assessment that Neanderthal did not produce a bone industry [1–6].

The discovery of more than 1,200 bone tools at the Neanderthal site of Chagyrskaya (Altai, Siberia, Russia) challenges this claim [7]. The cave deposits, accumulated in late Marine Isotope Stage (MIS) 4 or early MIS 3, contained 74 human remains and a lithic industry attesting to an occupation of the site by Neanderthals with cultural and genetic affinities to Micoquian groups from Central and Eastern Europe. Around 60,000–50,000 years BP, Neanderthals repeatedly came to the site during the early cold season to process the carcasses of hunted Bisons [8,9]. A technological and functional analysis of the faunal remains identified more than 1,000 retouchers and approximately 100 bone tools belonging to other functional categories. Although their manufacture involved percussion, with marginal use of scraping and abrasion, their number, diversity and recurrence lead to their consideration as an industry (systematic and organized production of a set of tools) [7].

To date, the Chagyrskaya bone tools provide the only example of a Neanderthal bone industry, at least, for which the authorship of AMH cannot be considered [10–12]. Bone tools have already been reported in Neanderthal sites but most of the time as isolated finds. A hundred kilometers away, such objects have been identified in contemporaneous levels of Denisova cave [13–15]. They may have been made, in part, by Neanderthals [9,16,17]. In addition to retouchers, some bone tools have also been reported in Crimean Micoquian assemblages from the sites of Prolom II [18], Kiik Koba [19], Karabi Tamchin [20], Zaskalnaya VI [21], and possibly Buran-Kaya III [22], as well as at Kůlna [23], in the Czech Republic, or at Salzgitter-Lebenstedt [24] in Germany. Outside the Micoquian zone, examples of Mousterian bone tools are even more numerous. Some have been found in Ukraine at the Molodova I site [25], in France, at the Bison Cave [26], at the sites of Combe-Grenal, La Ferrassie, La Quina [27–29], Pech-de-l'Azé I, Abri Peyrony [30], Montgaudier [31], Vaufrey [32], Noisetier [33], Canalettes [34], and Gatzarria [35], in Spain at the sites of Axlor [36], Bolomor [37], Arlanpe [38], Abric Romaní [39], and in Italy, at Fumane Cave [40]. These discoveries, most of them recently investigated, allow us to reconsider earlier ones which were potentially too readily dismissed such as those of La Quina [41], Ourbières [42], Tourtoirac [43], Néron [44], Pié-Lombart [45], Cuva Morín [46], Rigabe [47], Hauteroche or Bois-Roche [32].

The use of bone tools by Neanderthals is beginning to be discussed, if not accepted, because of this increasing number of reported cases. Most of these examples, only marginally shaped, mostly by percussion, have been identified without a precise methodological framework [32]. When identifications are not based on comparison with Upper Paleolithic bone tools, they are established from analytical criteria borrowed from lithic technology, or simply proposed by default when no other explanation can be provided. Doubts often remain because of a risk of confusion with forms resulting from natural alterations or butchery activities that may mimic, modify or erase traces of manufacture and use-wear traces [48–58]. As a consequence, a small number of items have been reported in each site, and Neanderthal bone tools, except retouchers, are still considered anecdotal. Because of their manufacturing techniques, bone is sometimes regarded as a substitute material [40,47,59,60]. This is likely due to a lack of understanding of the physical properties of bone and insufficient recognition of its mechanical qualities [32,61]. In any case, the diversity of published identification criteria and vocabulary make comparisons between sites difficult and prevents a more comprehensive consideration of a Neanderthal specific bone technology.

Based on the bibliographic data and the results obtained at the Chagyrskaya Cave, we hypothesize that the Neanderthals produced an original bone industry, different from the Upper Paleolithic standards, especially due to the predominant use of percussion. In other

words, the scattered finds mentioned above could be indicative of a more general phenomenon in Neanderthal technology, which has so far been underestimated due to a lack of appropriate methodological and conceptual frameworks of study. This could imply, on the one hand, that the previously found bone tools in various Middle Paleolithic contexts are only the most visible part of a production whose main components are yet to be identified and, on the other hand, that the Chagyrskaya bone industry is not a localized phenomenon resulting from a regional adaptation of eastern Neanderthals. To test this hypothesis, we applied to the bone assemblage of a Neanderthal settlement in western Eurasia the same techno-functional analysis as in Chagyrskaya, complemented with μCT imaging.

In 2019, the new excavation of Mousterian deposits at the Chez-Pinaud site (Jonzac, Charente-Maritime, France) provided the opportunity to reconsider the faunal assemblage of a western Neanderthal settlement. At this multilayered site, the Quina facies, corresponding to MIS 4, is characterized by a large bone-bed resulting from repeated seasonal occupations by groups that came during the cold season to process the carcasses of hunted animals on their migration routes [62]. In previous excavations, this bone-bed yielded numerous bone retouchers [63] indicating that bone was used as raw material for tool making. It is in this archaeological context, similar to that of Chagyrskaya in terms of activities and chronology, but geographically and culturally distinct [8,9], that we sought new evidence of a Neanderthal bone industry.

The Chez-Pinaud site was discovered in the late 1990s by E. Marchais. It is located on the right bank of the Seugne, a tributary of the Charente River (Fig 1a), at the foot of a cliff where a limestone quarrying area partially destroyed the deposit, during the 19th century, but also allowed its discovery [64]. It was excavated several times, in 1998–1999 and 2002–2003 by J. Airvaux, then between 2004 and 2007 by J. Jaubert and J.-J. Hublin. This fieldwork campaigns revealed the presence of Early Upper Paleolithic occupations and significant Middle Paleolithic levels attributed to three Mousterian facies: a Quina-type Mousterian, a denticulated Mousterian and a Mousterian of Acheulean Tradition [63].

In 2019, new excavations were initiated in the stratigraphic unit (US) 22 (Fig 1b), in close proximity to the 2004–2007 excavation area. US22 is composed of a yellow-brown sediment of clayey sands deposited by water and solifluction [63] (S1a and S1b Fig). Preliminary studies of the lithic industry from the 2019–2020 campaigns are in line with the results of previous excavations. Characteristic of the Quina facies, the lithic industry includes a high proportion of tools, mainly side scrapers, and their shaping and resharpening waste, while nuclei are rare. The contribution of exogenous raw materials and finished tools, in a context where high quality local flint is available, as well as the versatility of tools and the evidence of recycling, suggest a sparing use of lithic raw material [65]. This could stem from the high seasonal mobility of Quina groups [66].

The faunal spectrum of the US22 is largely dominated by reindeer remains (*Rangifer tarandus*), a marker of a cold, dry and open environment [67,68]. Individuals were killed in late fall and during the cold season and whole or sub-whole carcasses brought to the site for processing [62]. The large accumulation of bone (S1c Fig), which required a long period of time, and the consistency in activities, led to the consideration of the Chez-Pinaud site as a task-specific location (*sensu* Binford [69]) related to hunting arctic deer predation. Horse (*Equus caballus*) and bison (*Bison priscus*) complete the ungulate spectrum. Small mammals include fox, (*Vulpes vulpes* or *Alopex lagopus*), arctic hare (*Lepus timidus*) and marmot (*Marmota marmota*).

The spatial organization and state of preservation of artifacts indicate small-scale post-depositional movements of archaeological remains. Concentrations of lithic knapping waste, with refitting evidence, have been found in the US22 (see A. Delagnes [63]), as well as in situ bones anatomical connection from previous (see J.-B. Mallye, L. Niven, W. Rendu, T. E. Steel

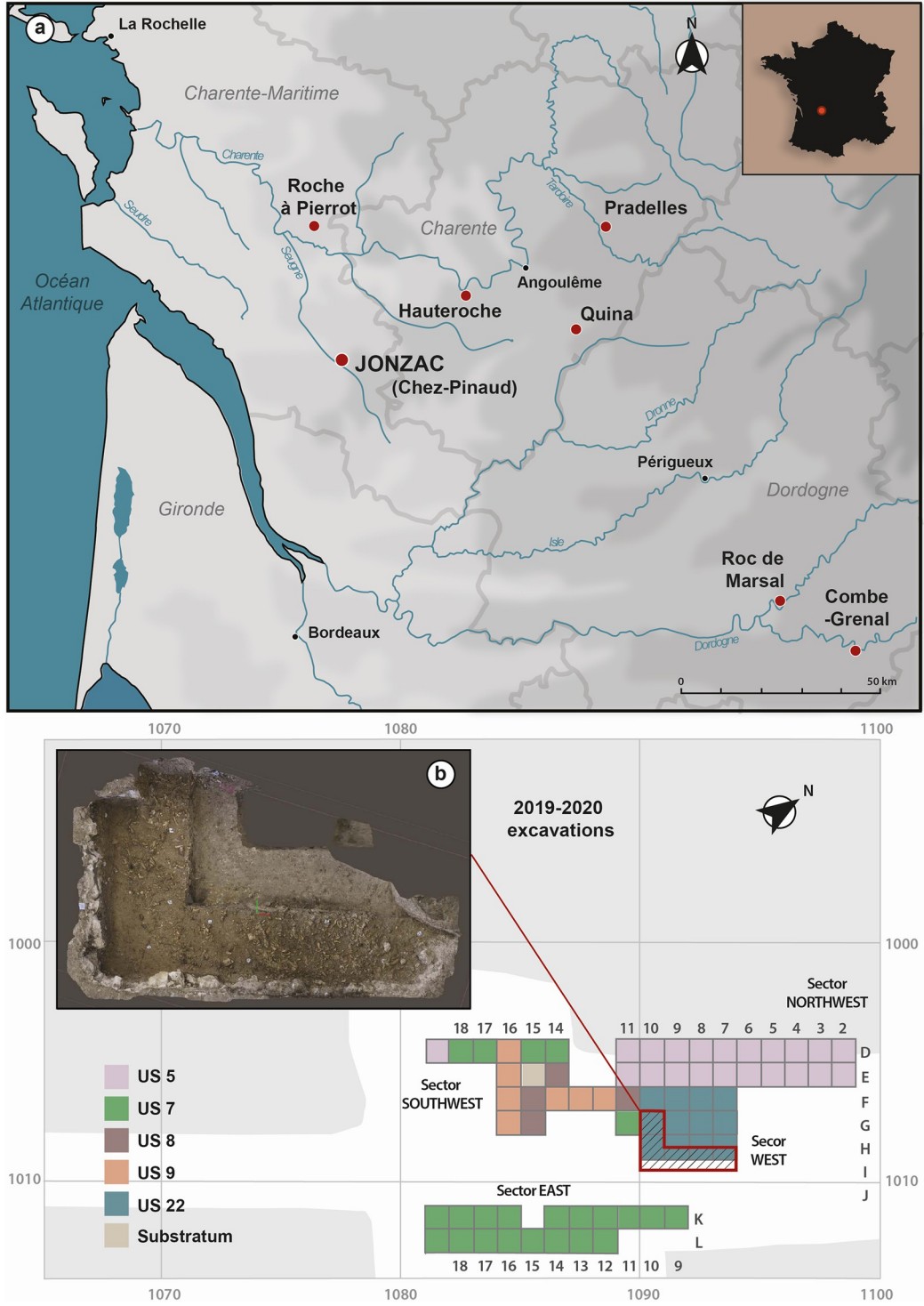

**Fig 1. The Chez-Pinaud site, Jonzac, France.** (a) location of the site and the main surrounding Quina site (geographical data republished from Géoportail.gouv.fr under a CC BY license, with permission from IGN, original copyright 2022). (b) 3D photogrammetric model of the excavated layer (photos and 3D processing: S. Shnaider). (c) layout of the excavation units (CAD: M. Baumann).

[63]) and on-going [70] excavation. There is almost no taphonomic abrasion (*sensu* Fernández-Jalvo and Andrews [71]) on the bones and the cortical surfaces are fully preserved on more than half of the faunal remains. Although visible on most pieces, weathering is limited to its first stages [72], indicating rapid sedimentation. The near absence of evidence of carnivore activity (absence of large carnivore remains and coprolites, extreme rarity of consumption traces < 1% NISP) shows that carnivores were only marginally involved in the accumulation and destruction of the bone assemblage [62].

US 22 was dated by Richter et al. [73], using thermoluminescence (TL) of heated flints, to $73 \pm 8$ ka (average of three individual ages). A new dating by single-grain Optically Stimulated Luminescence (OSL) on sedimentary quartz indicate an age around 60–65 ka (Guérin et al., in prep.).

## Materials and methods

The bone tools studied are those from the first two excavation campaigns (2019–2020) conducted by W. Rendu, K. Kolobova and S. Shnaider. They come exclusively from US22, attributed to the Quina Mousterian. They were identified following the sorting of all the faunal remains, i.e., little more than 3220 remains, stored in the Service départemental d'Archéologie of Charente-Maritime (France). The sorting was done with the naked eye, then pieces with possible evidence of manufacture or use were examined with a Nikon SMZ-1 stereoscopic microscope. The best-preserved tools have been documented with a Makroskop Wild M420 (apochromatic zoom 5.8x-35x), completed by a Canon EOS 1100D camera, and with an Olympus BH2 microscope equipped with semi-apochromatic objectives and interference contrast. Sequences of Makroskop shots with progressive focus shift were compiled with Helicon Focus software (HeliconSoft, Kharkiv) in order to extend the field depth over the entire framed area.

### Traceology

The methodological framework is that of traceology, as defined by S.A. Semenov, that is to say, taking into account both the traces of manufacture and use, at various scales [74,75]. In Western archaeology, these two complementary sides evolved separately through technology studies, which reconstruct the manufacturing *chaîne opératoire* of the artifacts from the shaping macro-traces, and through functional analyses, which infer their function from macro and micro use-wear. They are now at the heart of any lithic study, which is not yet the case for bone tools, whose remains are less numerous and more altered. Transfer from lithic to bone has been the most common methodological approach, both for the technological studies, with regard to sequencing of manufacturing operations [76–82], and for the use-wear analyses, with regard to the microscopic scale of examination [83–91]. However, it only concerned assemblages where the recognition of the artefacts, typologically established, did not constitute the issue of the study, and from particularly favorable preserving context, mostly of the Holocene period. It is now necessary to complete the referential frameworks by taking more account of the specificity of the bone material and by considering markers that are scarcely or not affected by surface alterations, i.e., macroscopic and/or internal.

### Taphonomic frame of reference

Our identification of organic (microorganism, animal, plant etc.) and inorganic (weathering, water circulation, sediment compaction etc.) taphonomic alterations is based on data outlined in publications [71,72,92–95]. We paid particular attention to the origin of bone fractures. The fractures considered as anthropogenic are those occurring on "fresh" bone, i.e., occurring shortly after the animal's death. They are characterized by a helicoidal shape and a slick surface

and form acute or obtuse angles with the cortical side. Post-depositional fragmentation generally occurs on "dry" bone, i.e., that has lost all or main part of its organic component. The fracture surfaces are straight, rough and form with the cortical side an angle close to 90˚ [61,96–100]. However, the transition from fresh to dry bone is a gradual and non-homogeneous phenomenon. Therefore, the proposed criteria do not allow for the recognition of an anthropogenic fracture on drying bone or a post-depositional fracture on fresh bone. In the archaeological material, the fracture surfaces often have mixed features, which implies reasoning in terms of trends according to the corpus studied and taking into account all available discriminating criteria (hammers impacts, general morphology, location, variability within the corpus, etc.) [101–105].

## Archaeological frame of reference

Our identification of the manufacture and use-wear traces is based on a comparison with published archaeological and experimental data as well as on our own experiments. The nature and organization of the traces observed on the archaeological material led to focus more specifically on morpho-functional categories already identified in Upper Paleolithic contexts and some Middle Paleolithic sites, those of: (1) bone retouchers, (2) beveled tools, (3) tools with retouched edges, and (4) tools with smoothed end. These categories are not defined with the same precision. For the bone retouchers, we know both their mode of utilization and the activity in which they were involved. They are light soft hammers for retouching lithic edges (S1 Text) [36,75,106,107]. For the beveled tools, the nature and localization of the traces refer only to a mode of use involving percussion, but the worked material and the task performed may vary (S2 Text) [35,108–110]. Bone fragments with intentional removals on one edge (S3 Text) [40,59,96,111] or with a technical blunt on one end (S4 Text) [7,30,55,88], are characterized only by a type of damage and its location relative to the main axis. For these categories, many hypotheses about their function can be put forward. It is therefore necessary, in order to complete the available data, to enlarge the experimental frame of reference.

## Experimental frame of reference

Experiments specifically dedicated to Mousterian bone tools were performed by a working group including French (PACEA, UMR 5199, University of Bordeaux), Russian (Institute of Archaeology and Ethnography, Siberian Branch of the Russian Academy of Sciences; ZooSCAn IRL 2013) and Belgian (Traceolab, University of Liege) researchers, among them experts in flintknapping and specialists trained to work with bones. These experiments refer to the entire *chaîne opératoire*: from the first fracturing step of the bones (S2a–S2d Fig) to the blanks shaping (S2e and S2f Fig) and until the tools-use on different worked materials and in task diversity in accordance with the archaeological contexts under study (S2g–S2l Fig).

For the purpose of this study, forty-five long bones of medium and large mammals were fractured: 16 femurs, 12 tibias, 11 humeri, 6 radio-ulna of cow (N = 41), red deer (N = 3) and *Equidae* (N = 1), adults (6 to 12 years old, N = 30), young adults (6 to 12 months old, N = 12) and juveniles (< 6 months old, N = 3). The long bones were split with a hard hammer (quartzite pebble of 1 to 2.5 kg) on an anvil (in situ limestone bloc or large quartzite pebble). Only the location of the impact points varied from one bone to another. Other types of bone, such as ribs and vestigial metapodial (red deer and *Equidae*), were also used without fracturing step, or after a first step reduction limited to a transverse fracturing. Twenty fractured bones have been set aside for analyzing the variability of the percussion marks and the bone fragments morphometry. They correspond to 156 flakes (excluding chips of less than 3 cm length and

epiphyses), i.e., about 7–8 blanks per bone, with an average length of 10.28 cm (L. max. = 19.98 cm, L. min = 4.08 cm) and an average width of 4.3 cm (W. max = 9.88 cm, W. min = 1.46 cm).

The blanks shaping consisted mainly of a second fracturing step by direct percussion for sizing, gripping and/or the active edge setting, i.e., modification of the edge angulation and/or delineation. We tested the efficiency of different hammers: soft organic (antler, Juniperus communis), soft mineral (fine sandstone, limestone, schist), hard mineral (quartzite) depending on the configuration of the striking platform, the size, morphology and axis of the removals to be made. Tests were carried out to evaluate the bone response according to its state of freshness, i.e., fresh (a few days after the death of the animal) and dry (loss of fatty and wet components) and our results were compared with those of previous experiments [7,32,40,59,96,111–114].

Experiments in bone knapping converge on similar observations: (1) bone responds favorably to percussion, free or on anvil, (2) fresh bone responds to knapping better than dry bone, i.e., better control of the removals morphology, (3) all hammer types (vegetal, animal and mineral) allow retouching, (4) the more acute the striking angle, the more efficient the knapping; it is generally between 40˚ and 60˚ and should not exceed 80˚, (5) removals are easier to be performed when the percussion is applied parallel to the direction of the bone fibers than perpendicularly, (6) the bone flakes have a butt, but the associated traces such as the bulb, bulb scars, hackles, or ripples, are not systematic and not as clearly observable as on the lithic. The results diverge about the removals morphology according to the type of hammer. In our experiments, a distinction can be made between the hard mineral hammers (quartzite) and the soft hammers (wood, antler, soft sandstone). The latter allow to perform, under similar conditions, more invasive and scaled low angle removals than the hard hammers. We also nuance previous results on the response of dry bone to knapping. The more homogeneous the material is, the easier it is for the shock wave to propagate. After the death of the animal, the bone gradually loses its organic mass, starting from the outer layers, through internal mechanisms of cell death and external mechanisms of exchange with the environment. This loss of homogeneity leads to irregularities in the morphology of the removals. However, once completely dry, as long as it has little or no weathering damage, the bone recovers its homogeneity. More rigid, it is particularly suitable to knapping.

Our bone tools were used for flint knapping (e.g., scrapers shaping), woodworking (e.g., handles manufacture), plant working (e.g., herbaceous harvesting), skin working (e.g., proto-tanning), ice working (e.g., fishing hole), butchery (e.g., meat cutting), and soil working (e.g., digging up roots).

## Mechanical bone properties

Bone is a complex matter composed mainly of an organic matrix of collagen fibers and an inorganic matrix of hydroxyapatite crystals, organized in lamellae along the anatomical axis. Changes in fiber orientation from one lamella to another form a solid and relatively homogeneous frame, reinforced by the prevalence of the mineral component. At the macrostructural scale, bone has viscoelastic properties and transverse isotropic reaction. It can absorb energy by reversible deformation, fix a permanent deformation, and break if the stress exceeds its resistance [61,96,115–118]. Its deformation capacity also depends on the rate of the applied stress. The higher the latter, the higher the stiffness response of the bone. In addition, the progressive dehydration of the matter, after the animal death, increases its hardness [119]. Bone can therefore, depending on the stress application conditions and the matter freshness, behave as a rather ductile or rather brittle material. It shares characteristics with conchoidal fracture materials (development of a percussion cone below the impact area) [95,120], and as the latter, responds to stresses in a predictable manner.

### Internal traceology

The plastic deformation capacity of bone, that lithic does not have, opens an additional research field on deformations and cracks localized in the stress area, which may affect not only the bone surface but also the underlying internal structure and are likely to supply diagnostic criteria for tool recognition. Among the imaging techniques providing access to the internal structure of solid matters, microtomography [121] is particularly suitable for the observation of damage and micro-damage into the cortical tissue [122–128]. It is a non-destructive imaging technique based on X-ray slices recombined for getting a 3D model of the scanned object, according to the principle of Computerized Tomography [129,130]. Used mainly in the medical field, this technique is still little involved in the studies of archaeological bone artifacts [131–137]. μCT could enable the implementation of a method of "internal traceology" for supplementing surface information or for compensating for it when it is altered.

We performed μCT capture, without prior treatment of the samples, on 9 bone tools from the Chez-Pinaud site (S1–S9 Files) and 5 bone tools from our experiments, getting a total of 25 scans (S1 Table). The archaeological bone tools were chosen for their particularly good state of preservation, and to cover the morpho-functional diversity. The scanner chamber dimensions (260 mm in diameter and 420 mm in length) allowed a complete capture of each specimen at voxel sizes between 98 μm and 26 μm while scans of ROI (region of interest) were performed at resolutions between 33 μm and 5 μm (minimum values fixed by the volume to be covered). The μCT operated at a voltage of 110 to 120 kV and an intensity of 110 to 200 μA. The X-ray beam was filtered by a 0.1 mm thick copper plate.

We used a General Electric V|Tome|xs dual source microscanner (PACEA laboratory, University of Bordeaux, France). Volume reconstruction and visualization were performed with AVIZO 7.1 software (Thermo Fisher Scientific, Waltham).

## Results

### General features

Within the faunal remains from 2019–2020 campaigns, we identified a total of 103 bone tools. The feasibility of use-wear analysis is constrained by a loss of material on the surface of the bone due to exfoliation phenomena (delamination of the outermost cortical layers). Longitudinal cracks from the weathering, combined with the weight and nature of the sediment and with the passage of the quarry machinery, probably caused much of the fragmentation of the bone remains. Only 15% of the tools are complete or sub-complete, i.e., free of post-depositional or recent breakage.

Of the 103 bone tools, a majority are marked by scores areas (*sensu* Mallye et al. [138]) localized on the cortical face (N = 91), some have removals at one or both ends of the main axis (anatomical axis; N = 10), and/or removals on the cortical and/or medullary surface of a lateral edge (N = 7) and/or a smoothed area, more or less extensive, at one end (N = 2). The main modifications, which changed the volume of the blanks, are always associated with one or more marginal modifications such as chipping, striations, compactions, rounding etc. The main modifications correspond respectively to the following morpho-functional categories: (1) bone retouchers, (2) beveled tools, (3) retouched tools, (4) smooth-ended tools (Table 1; Fig 2). More than 88% of the blanks were used as retouchers, and 81% exclusively in this way. Multiple-use tools accounted for 9% of the total. The most common association is a bone retoucher with a beveled end, followed by a bone retoucher with a lateral retouched edge. To date, there are no tools that combine the four major modifications.

**Table 1. Bone tools, Chez-Pinaud site, 2019–2020 excavations.**

| Number of tools | Type | Number of blanks |
|---|---|---|
| 103 | Bone retouchers | 91 |
| | Beveled tools | 10 |
| | Retouched tools | 7 |
| | Smooth-ended tools | 2 |

Bone blanks number and count of active area per tools category.

All tools are made from bone, except for a horse incisor used as retoucher [139]. The taxa used in their manufacture are the same as those hunted and brought to the site: reindeer (34%), horse (26%) and *Bos*/bison (5%), but with different proportions (Fig 3d). Bones of large ungulates are almost twice as numerous among the tools as among the faunal remains (Fig 3a and 3b). While bone retouchers are made equally from large (53.8%) and medium (46.2%) ungulate bones, multiple tools are always made from large ungulate bones. The blanks are mainly flakes of long bones (87%): tibia (42.6%), humerus (17%), femurs (17%), metapods (12.8%) and radio-ulna (10.6%), for anatomically determined pieces. Ribs (8.7%), phalanges (N = 2), coxal (N = 1) and scapula (N = 1) fragments complete the corpus.

The blanks were split by direct percussion. Percussion marks and notches, sometimes associated with microcracks and/or secondary splinters, are visible on, at least, 30% of the tools. Fracture morphologies show that the bones were fractured in a fresh state. The lengths of the complete and sub-complete tools range from 5.6 cm to 15.4 cm (L. avg. = 9.1 cm) for widths between 1.8 cm and 5.1 cm (W. avg. = 3 cm). Thicknesses, rarely altered, were recorded on all tools. They vary from 0.1 cm to 1.41 cm (Th. avg. = 0.7 cm). The general characteristics of the blanks used to manufacture the tools are comparable to those of the faunal remains of the same assemblage. These faunal remains show that all long bones types were fractured, with approximately 17% percussion marks, while the fresh fracture rate reaches 76% of long bones. Eighty percent of the bone fragments correspond to a quarter or less of the shaft (reindeer < 5–6 cm; horse/bison < 7–8 cm), and 7% to between a quarter and half of the shaft (5–6 cm < reindeer < 13–14 cm; 7–8 cm < horse/bison < 16–17 cm). The lengths of the complete bone tools from medium size ungulates of the 2019–2020 excavation fall within this latter range (8 cm < medium size ungulates < 12 cm), while the lengths of the complete bone tools from large size ungulates cover the two sets (5 cm < large size ungulates < 16 cm).

## Bone retouchers

**Morphometry.**    Retouchers are the most frequently found bone tools in Middle Paleolithic contexts (S1 Text) [36,37,60,138,140–145]. US22 already had yielded nearly 510 during previous excavations; these retouchers were described and analyzed (Airvaux excavations: 202 specimens [146], Jaubert-Hublin excavations: 307 specimens [147]). The 91 bone retouchers from the new excavations show the same characteristics, indicating a continuity in the activities and choices made by the Quina in the handling of these tools. The blanks are 54% from large size ungulates (55% in Airvaux excavations and 47% in Jaubert-Hublin excavations). The tibia (22%) is the most used bone, but it is also the easiest to identify so our estimation might be biased. Flat bones represent 8% of the corpus and the use of short bones is anecdotal (2 phalanges). The thickness of the retouchers is directly correlated with the bone used, with an average value of 0.86 cm for large size ungulates and 0.45 cm for medium size ungulates. The length of the most complete specimens ranges from 5 cm to 9 cm, with a width of 2 to 3 cm. However,

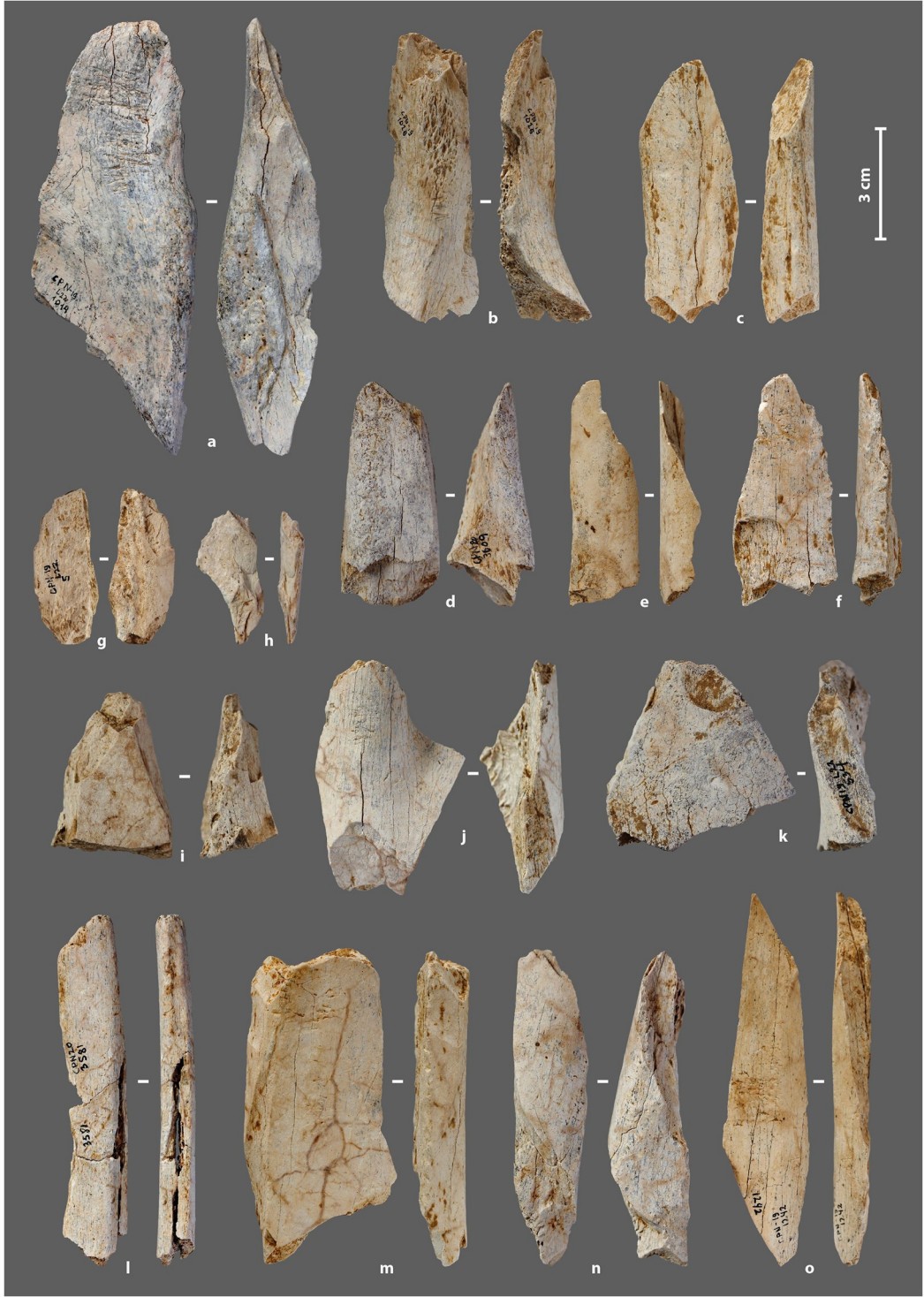

**Fig 2. Bone tools from the Chez-Pinaud site, 2019–2020 excavations.** (a–d), (i–k) beveled tools. (a, g–h, e–f, n) retouched tools. (a, f, j, m–o) bone retouchers. (l) smoothed-end tool (photo: M. Baumann).

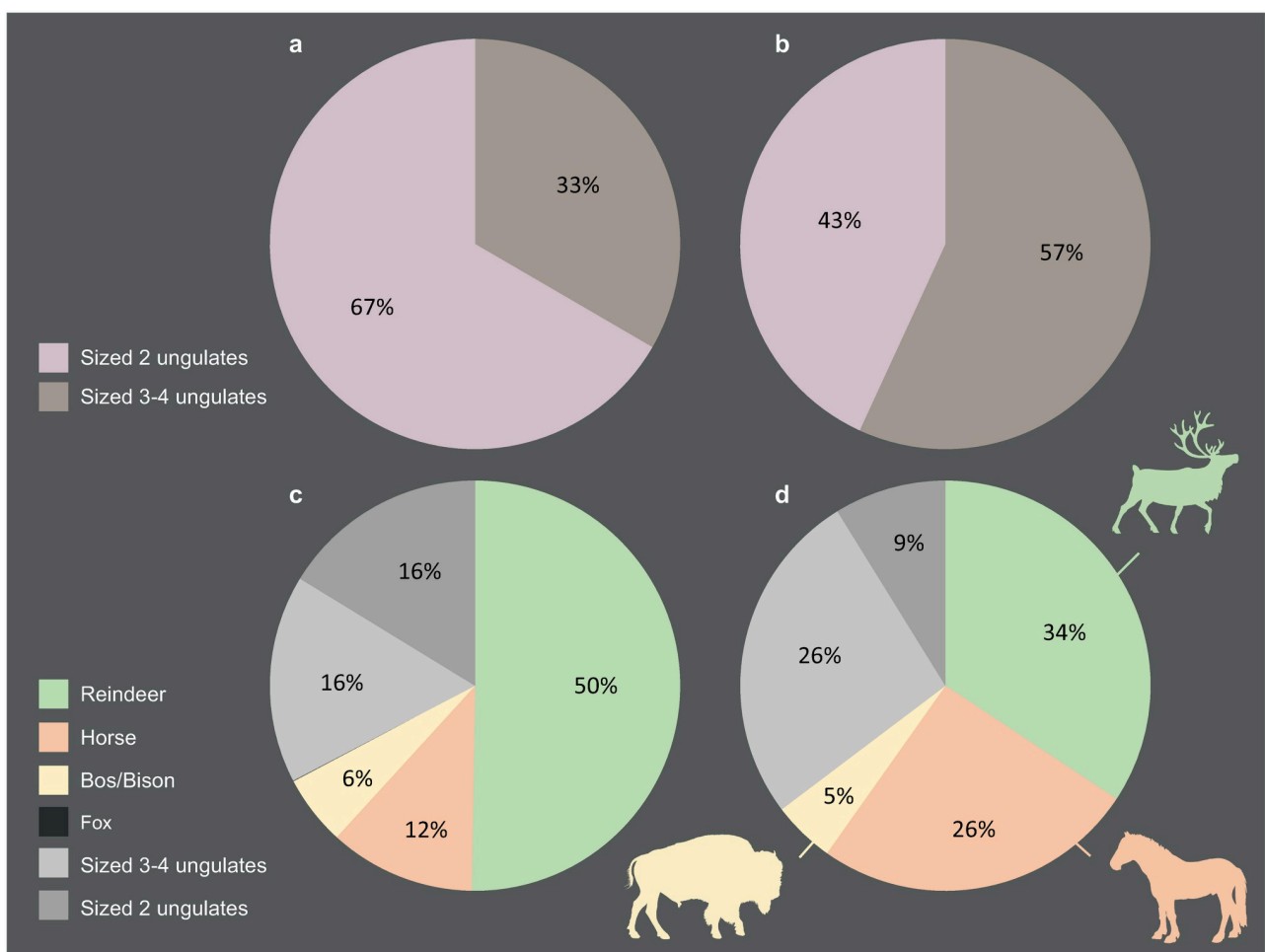

**Fig 3. Species spectrums, Chez-Pinaud site, 2019–2020 excavations.** (a-c) faunal remains. (b-d) bone tools (CAD: M. Baumann).

the range of values is much smaller in the retouchers made from medium size ungulate bones than in those made from large ungulates, which may reflect stricter selection to achieve the dimensions required by the tools. Selection was also observed in Quina-type Mousterian layers at Les Pradelles, a secondary butchery site, where reindeer accounted for 98% of the faunal remains and provided 95% of the blank retouchers [144]. At the Chez-Pinaud site, the average length of medium size ungulate retouchers (8.5 cm) is greater than those from Les Pradelles (7.3 cm). The higher proportion of large size ungulate remains at Chez-Pinaud likely provided enough suitable blanks to make a stronger selection.

**External modifications.** On our 91 bone retouchers, scores areas are mainly single (73%), sometimes double (24%; Fig 4d), more rarely triple (3%), with prior scraping of the striking zone in 23% (Fig 4c). Their density varies from a few scores (3–4) to the superposition of several dozen (Fig 4i), the latter being less frequent. The scores are oriented perpendicular to the main axis with slight variations of obliquity (Fig 4a). They are linear, more or less rough, and sometimes associated with small chip removals (Fig 4b), that, according to Mallye et al. [138] indicates the use of dry bone fragments, or at least, fragments which have lost some of their moist, fatty surface component (Fig 4b).

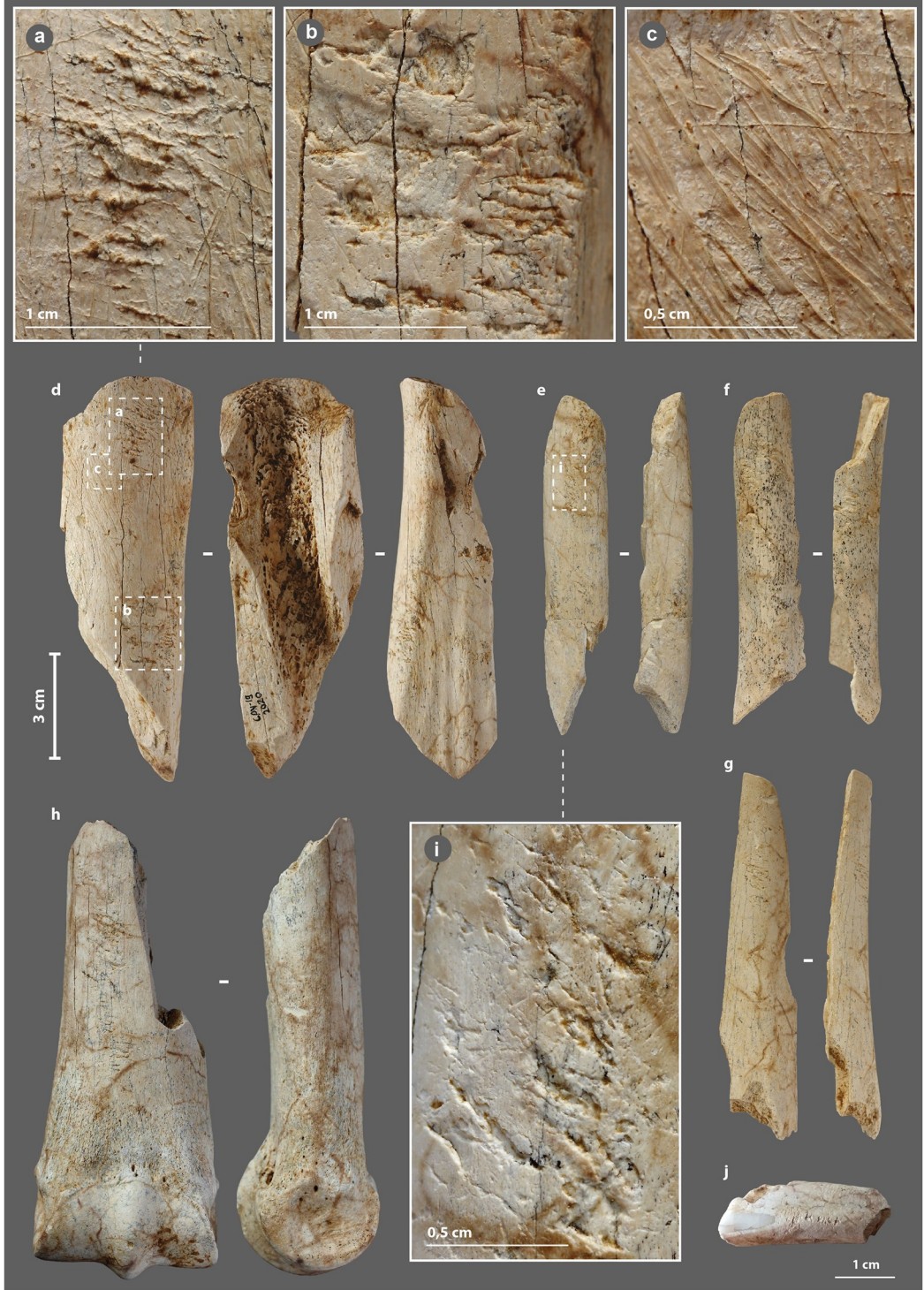

**Fig 4. Bone retouchers from the Chez-Pinaud site, 2019–2020 excavations.** (a) scores orientation. (b) scores associated with small chip removals. (c) scraping of the striking area. (d) on horse tibia, n˚2020. (e–g) on reindeer diaphysis, n˚3727, 3396 and 3320. (h) on horse metapodial, n˚3587. (i) dense scores area. (j) on horse incisor, n° 4061 (photo: M. Baumann except j, S. Renou).

At the Les Pradelles site, three groups of retouchers were distinguished based on tools size and score intensity: those, (1) dedicated to shaping and resharpening Quina scrapers, (2) corresponding to shaping and resharpening other tools and, (3) used occasionally for resharpening various active edges [144]. This proposal highlights that the variability of retouchers, often understood as the result of opportunistic use of non-specialized tools, may be more a matter of a "case by case" selection, adapted to the retouch to be performed. At Chez-Pinaud, groups of retouchers stand out, such as the small elongated specimens on the diaphysis of reindeer (Fig 4e–4g) with two short, well-marked scores areas (Fig 4i). The types of lithic edges retouch depend on multiple factors, including the weight of the bone retoucher, its handling capabilities, its trajectory or the energy transmitted [148,149]. The use of small elongated flakes sets a range of possibilities that is not that of semi-complete metapod specimens (Fig 4h), itself quite different from that of ribs or horse incisor (Fig 4j).

**Internal damage.**   The mode of use of a bone retoucher can also be revealed by its internal damage. The retouching of a lithic edge with a bone retoucher is the result of a reciprocal effect of one another, combining percussion and cutting. The lithic edge penetrates the bone by cutting and separating the fibers, while the energy transfer takes place at the bottom of the cut when maximum penetration is reached. The percussion does not only initiate a crack on the impact lithic edge but also in the bone material. The propagation of the latter should depend on many parameters related to the applied stress (dynamics of the blow, position and handling of the retoucher, kinematics, etc.) and the state and structure of the bone (anatomical characteristics, location of the active zone, humidity, temperature, etc.) [61,93,96,98,99]. The development of use cracks and micro-cracks into bone tools has already been demonstrated with a reindeer antler hammer [134] and some projectile points [132,136]. We therefore sought to investigate this type of damage through μCT imaging of three experimental (S2 Table) and three archaeological bone retouchers (S5–S7 Files).

Cracks and micro-cracks were detected on our three experimental specimens. They are frequent but not systematic and extend in the direct extension of the bottom of the scores. They are mainly simple (Fig 5e), but can also be complex, with one or more secondary cracks from a main crack (Fig 5c). They always follow a curved path with a tendency to reach the bone surface. Cracks can lead to partial chip detachment (Fig 5b), while the formation of a first crack promotes shrinkage if a second crack develops nearby (Fig 5d). All cracks and micro-cracks have the same orientation, in accordance with the regularity of the flintknapper's gesture. They can therefore provide information about the direction of the movement and possible changes in the position of the bone retoucher or lithic edge during the knapping process. At our scale of analysis, simple visual examination does not detect significant differences in crack length, depth, or delineation, based on the freshness states of the experimental bone retouchers.

Internal visualization of our archaeological bone retouchers is a delicate task. First, because we are dealing with fossilized material, i.e., with a large mineral component. The density of the material is greater and the grey levels recorded by the μCT no longer reflect, or only partially reflect, the original structure of the bone. Secondly, because the bones from Chez-Pinaud are altered by cracks from weathering that could mimic the technical cracks. Weathering cracks develop along the natural lines of weakness of the bones, i.e., appearing perpendicular to the surface, when the bone is observed along its main axis (Fig 5j) and parallel to the surface, when observed in cross-section. Cracks with similar characteristics as the damage caused by the use of experimental retouchers are visible under the scores area at 30–20 μm resolution (Fig 5g–5j). These initial observations validate our hypothesis of the formation of use cracks below the surface of the retouchers scores areas and the possibility of their detection in the archaeological material.

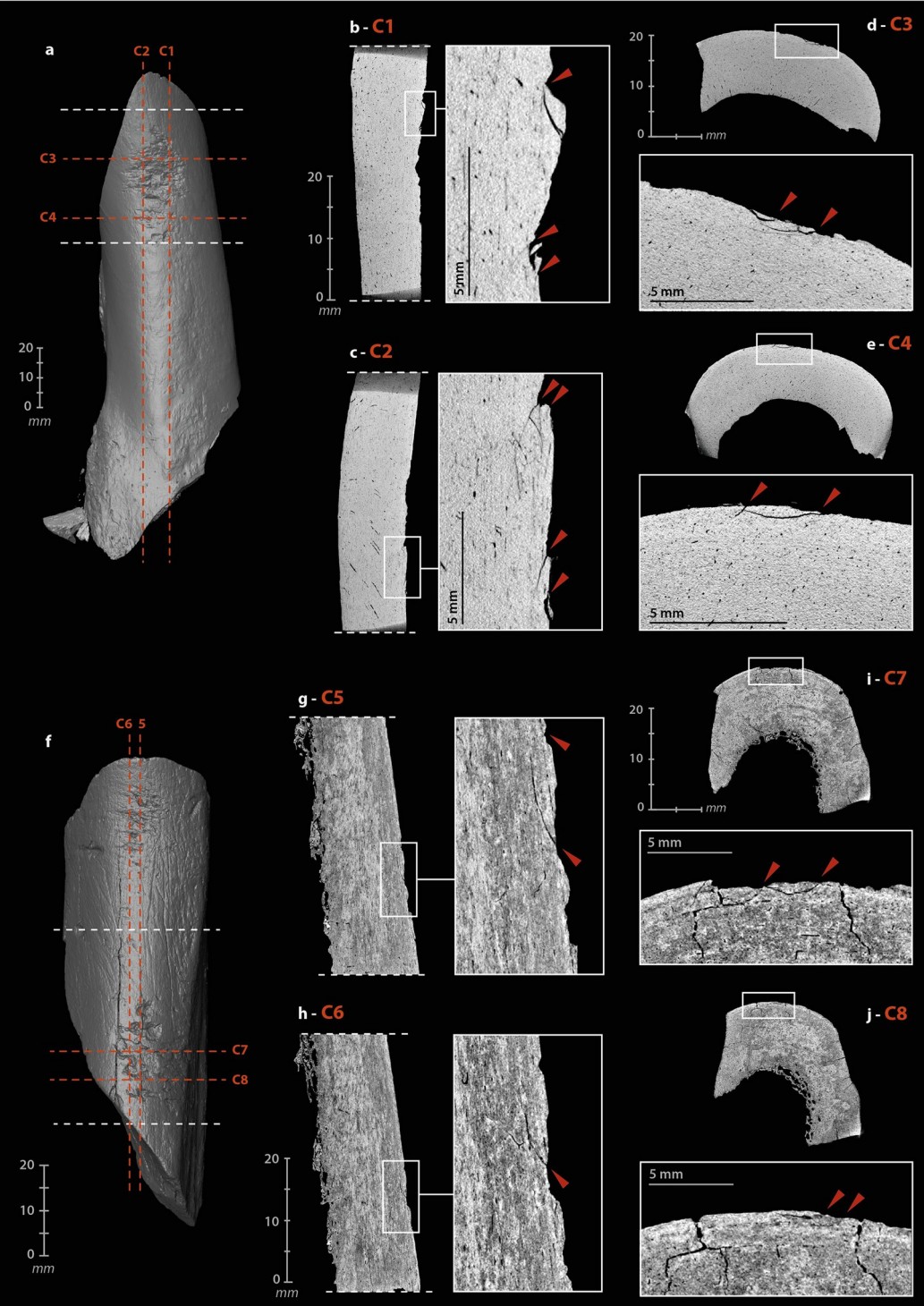

**Fig 5. μCT imaging of bone retouchers.** (a) 3D reconstruction of an experimental sample from a cow tibia. (b–c) cracks from use under the scores area, on sagittal cross-sections. (d–e) cracks from use under the scores area, on transverse cross-section. (f) 3D reconstruction of an archaeological sample from horse tibia, n˚2020. (g–h) cracks from use under the scores area, on sagittal cross-section. (i–j) cracks from use under the scores area, on transverse cross-section (μCT image processing [IP] and CAD: M. Baumann).

## Beveled tools

**Morphometry.** The morpho-functional category of beveled tools is well documented in Upper Paleolithic and Neolithic contexts (S2 Text) [35,85,88,108–110,150–152]. At Chez-Pinaud, the seven beveled tools range in length from 4.46 cm to 11.84 cm (L. avg. = 6.89 cm), for widths from 2.51 cm to 5.11 cm (W. avg. = 3.51 cm) and cortical tissue thicknesses from 0.56 cm to 1.36 cm (Th. avg. = 0.99 cm). Six of them have lost their original length. Two have a post-depositional and a recent breakage, respectively, while the other four are shortened by a transverse fracture at the proximal extremity. The latter have the attributes of a rapid fracture on fresh bone, posterior to the blank fracture. Their hinge morphology (fracture initiation from the upper side; Fig 6g) or sawtooth morphology (fracture initiation from one of the lateral sides; Fig 7g) [79,82,153,154] and limited extent are characteristic of bending stress near the fracture line. They may be the result of tools use. In one case, the transverse fracture is covered by a striking zone (Fig 6f), indicating that the tool was reused after its deterioration. This striking zone demonstrates that the beveled tool was still efficient despite its relatively short length (L. = 6.5 cm), which is less than the average length of all specimens.

The beveled ends of our archeological specimens are formed by the cortical surface and one of the fracture surfaces caused by the blank production at an angle of about 30˚ to 70˚. This particular morphology must therefore have been obtained during the first stage of fracturing; a result that is far from systematic given the morphological diversity of bone fragment extremities within the faunal remains of US22. Our experimental fracturing on fresh bone also produced limited fragments with a morphology suitable for direct use as a bevel. Several non-exclusive hypotheses can be put forward: (1) a control, during the first stage of fracturing, of the parameters required to obtain blanks with convex beveled ends of acute angle, (2) an anticipation by the selection and setting aside of suitable blanks following the processing of the carcasses, (3) a shaping by percussion of the active end posterior to the first fracturing stage. It is also necessary that the edge to be shaped presents a favorable knapping angle. Two pieces could illustrate this last case.

**External modifications.** The distal extremity of the shortest piece (Fig 8a) shows two generations of 3 or 4 longitudinal lamellar removals, at the junction between the cortical surface and a lateral fracture. Their location, regularity and organization, seem to refer to the burin blow technique, i.e., to intentional removals. As with a lithic burin, the objective could be the shaping of the active part and/or a production of bladelets. Experimentally, the removal of small elongated flakes along its main axis does not raise any particular difficulty, provided that the flaking angle is favorable. In this axis, the propagation of the shock is facilitated by the absence of curvature and the longitudinal macrostructural organization of the bone fibers. In all cases, the piece was used. Its distal extremity is blunt and chipped (Fig 8b). On the larger specimen (Fig 8e), the medullary surface of the bevel has a large and low-angled removal initiated from a lateral edge (Fig 8h). The latter is posterior to the first stage of fracturing of the blank and partially covered by use-wear traces (chipping). Given the initial thickness of the blank edge, it could correspond to a thinning form the bevel.

The bevels are bifacial (N = 5) or unifacial (N = 2), on the cortical and medullary surfaces (N = 6; Fig 7a) or on a lateral side (N = 1; Fig 7d). The lateral beveled tool shows a proximal fracture that is also lateral, in agreement with the axis of the stress exerted on the tool. All specimens show blunting of the cutting edge (Fig 7c), sometimes associated with visible compaction by flattening and wrinkling of the protruding areas. In 6 cases out of 7, the blunting is also associated with bifacial chipping (Fig 6a and 6b), more advanced on one face of the bevel (Fig 8g). We count one to four generations of small removals in the main axis, with a slight variation in their orientation. The bevel without a chipped end is marked by a bending fracture topped with a pronounced bluntness on the front and medullary surface (Fig 6e).

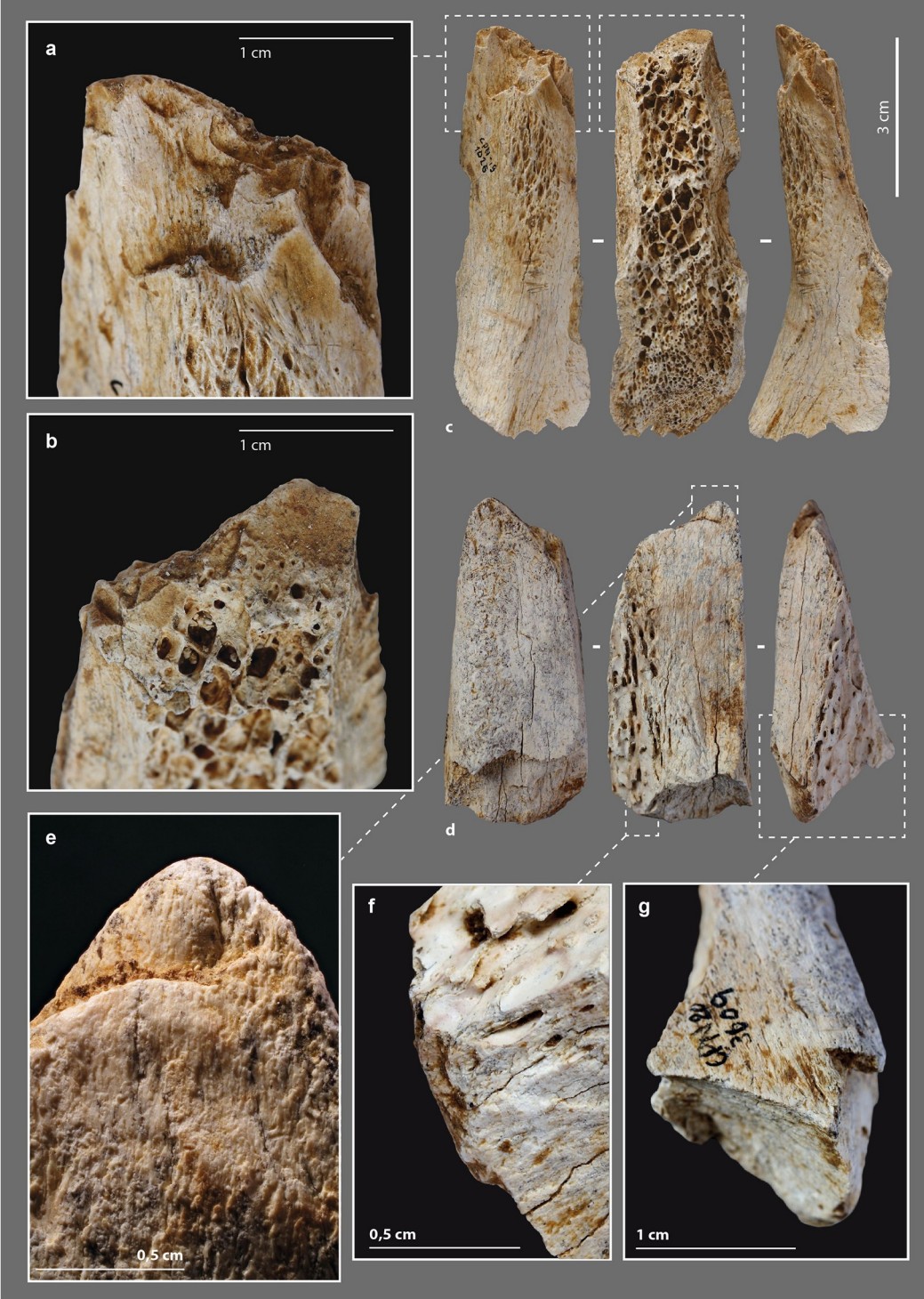

**Fig 6. Beveled tools from the Chez-Pinaud site, 2019–2020 excavations.** (a) chipping on the cortical surface at the distal end. (b) chipping on the medullary surface at the distal end. (c) on medium sized ungulate diaphysis, n˚1026. (d) on large size ungulate diaphysis, n˚3609. (e) blunt on the medullary surface at the distal end. (f) striking surface at the proximal end. (g) hinge fracture at the proximal end (photo: M. Baumann).

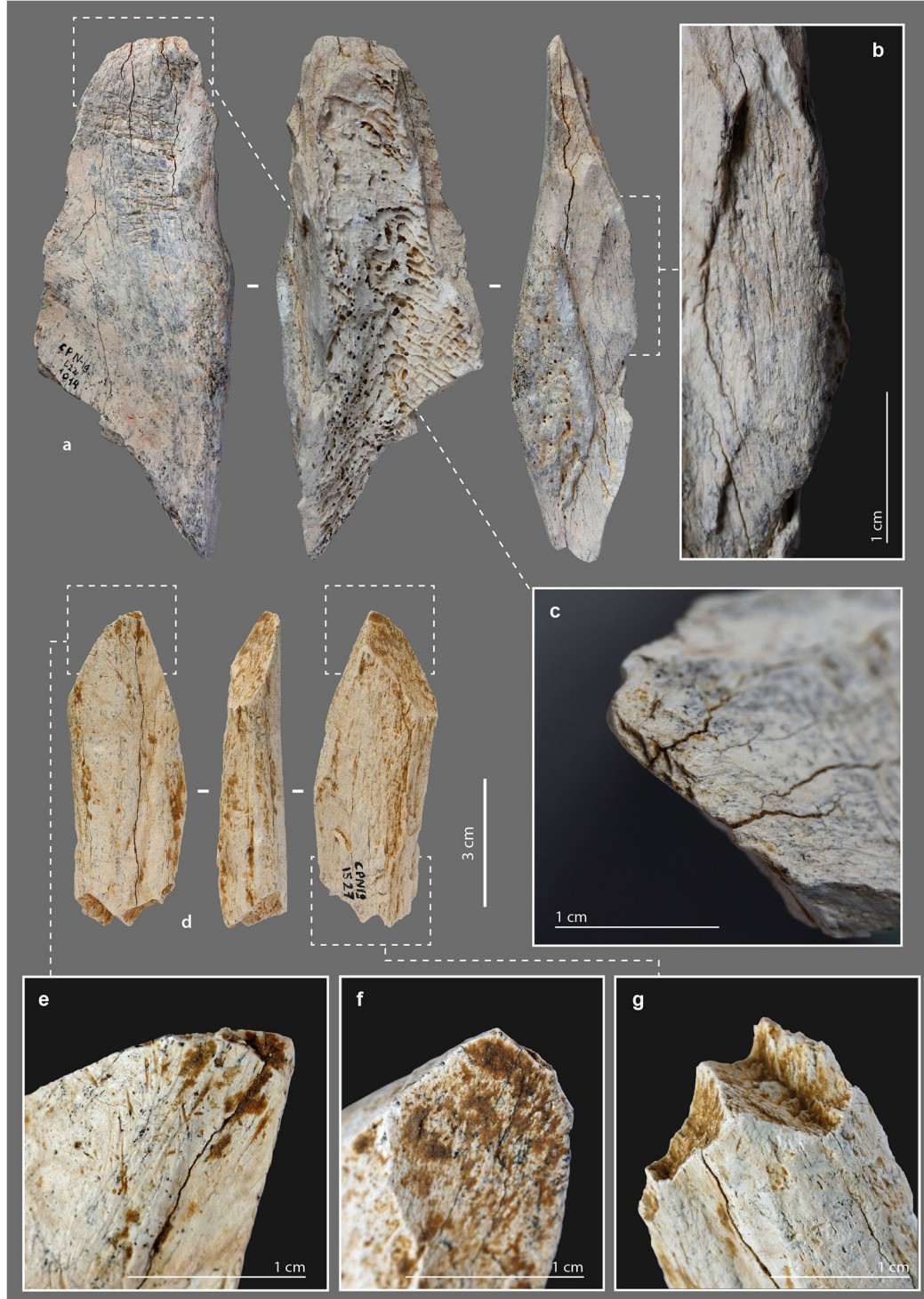

**Fig 7. Beveled tools from the Chez-Pinaud site, 2019–2020 excavations.** (a) on horse humerus, n°1014. (b) edge retouch on the cortical surface. (c) chipping partially covered by a blunt on the cortical surface at the distal end. (d) on bison metatarsal, n°1527. (e) chipping on the cortical surface at the distal end. (f) chipping on the medullary surface at the distal end. (g) sawtooth fracture at the proximal end (photo: M. Baumann).

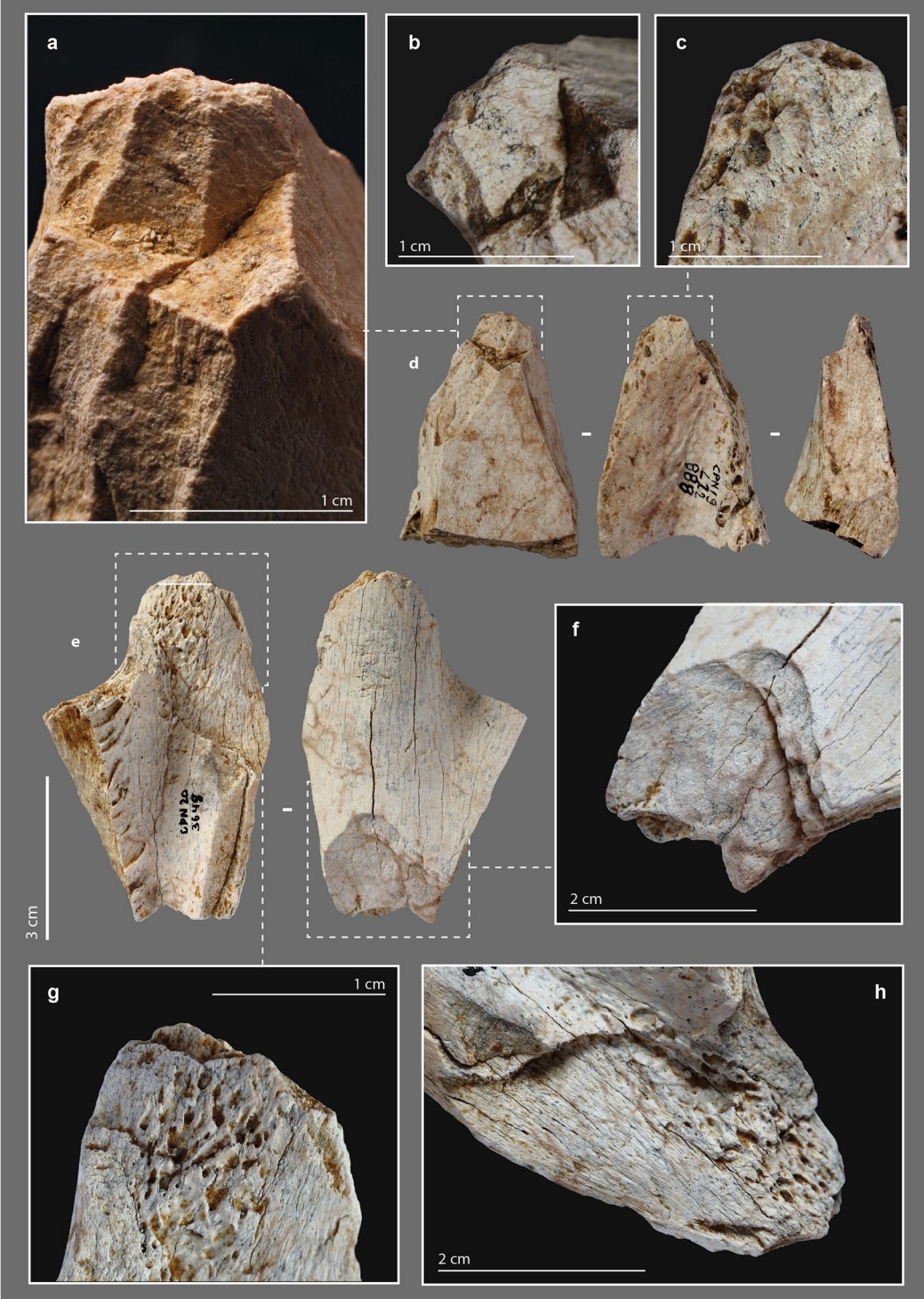

**Fig 8. Beveled tools from the Chez-Pinaud site, 2019–2020 excavations.** (a) set of burin-like removals. (b) blunt over removals at the distal end. (c) chipping on the medullary surface at the distal end. (d) on medium size ungulate diaphysis, n˚888. (e) on large size ungulate diaphysis, n˚3648. (f) hinge fracture at the proximal end. (g) chipping on the medullary surface at the distal end. (h) large and low-angle removal initiated from the lateral edge (photo: M. Baumann except a, H. Plisson).

**Internal damage.** The operating process of the beveled tools involves percussion, which motivates the search for internal cracks and micro-cracks. The cutting edge of a bevel is only effective on a raw material of lesser hardness. Nevertheless, depending on this hardness, the contact between the two can alter the bone. Here, the stress is applied longitudinally to the bone fibers. To test for use-related internal damage in beveled tools, we scanned two experimental (S2 Table) and four archaeological specimens (S3–S5 and S9 Files).

We did not identify any evidence of bone fiber deformation in the experimental specimens. However, macroscopic cracks developed in both tools. These were cracks extending a few millimeters under the surface, visible on sagittal and frontal μCT cross-sections at the distal ends. These cracks are direct extensions of removals produced during use (Fig 9b, A). The short cracks located around the striking platform (which may subsequently cause chipping of the periphery) are different from those that develop in the medial part. The latter penetrate only one or two millimeters along the main axis, but extend further in width (Fig 9e, E). We noted one multiple crack (Fig 9d, E). In one specimen, three cracks also formed starting from the medullary surface of the distal part, below the area in contact with the worked material. They follow the same direction for a few millimeters (Fig 9b, B–D). If the cracks located directly under the active extremity can be related to shocks received along the main axis, i.e., characterized by a dominant compressive stress, those located further on the periphery could result from repeated bending stresses, i.e., correspond to fatigue cracks.

For the archaeological beveled tools, the selected anatomical fragments are close to the epiphyses. The organization of bone fibers is more complex than in the diaphysis. Unlike experimental beveled tools, where the main axis of the tool is the same as the anatomical axis, archaeological beveled tools have their active extremity positioned obliquely to the anatomical axis. The cracks on the medullary surface, located on the periphery below the distal extremities, cannot be directly compared to those of the experimental specimens, as they follow the natural lines of weakness in the bone structure. We can only point out that the cracks concentrate at the active extremity (Fig 9h, A) and that there may be a connection between the technical micro-cracks and the development of taphonomic macro-cracks. Here, if the modifications were solely taphonomic, their distribution would be uniform, but parts of the beveled edges still show sharp and acute zones (Fig 9j), while others are chipped (Fig 9l), compressed (Fig 9k) or blunted (Fig 9m).

## Retouched tools

**Morphometry.** This category includes all tools in which one edge has been, according to the pattern of removals on the bone blank, intentionally retouched (S3 Text) [7,27,28,32,36,40,155–157]. The sizes of the retouched tools from Chez-Pinaud are quite heterogeneous, but the average cortical thickness, at 0.9 cm (σ = 0.2), is among the highest for all tools. This could be related to the need for sufficient thickness to knap the blank. The retouched tool lengths are probably not the original ones. Specimens without recent or postdepositional breakage (N = 4) have an average length of 8.7 cm. They were fractured, at least at one extremity, subsequent to primary stage fracturing to obtain the blank (Fig 10c).

**External modifications.** Two pieces of our archeological specimens show, at one extremity, the beginning of edge convergence and a convex front like those of the beveled tools. Recent and post-depositional breaks prevent from asserting the presence of a bevel and/or damage typical of this class of tool. However, on both specimens, the preserved part is covered with a blunt that is too localized to be taphonomic (Fig 10b). In addition to this localized blunting, they are both also retouched. On one of them (Fig 10a), the removals are located at a lateral edge of the distal extremity and were performed in two stages. The first removals performed,

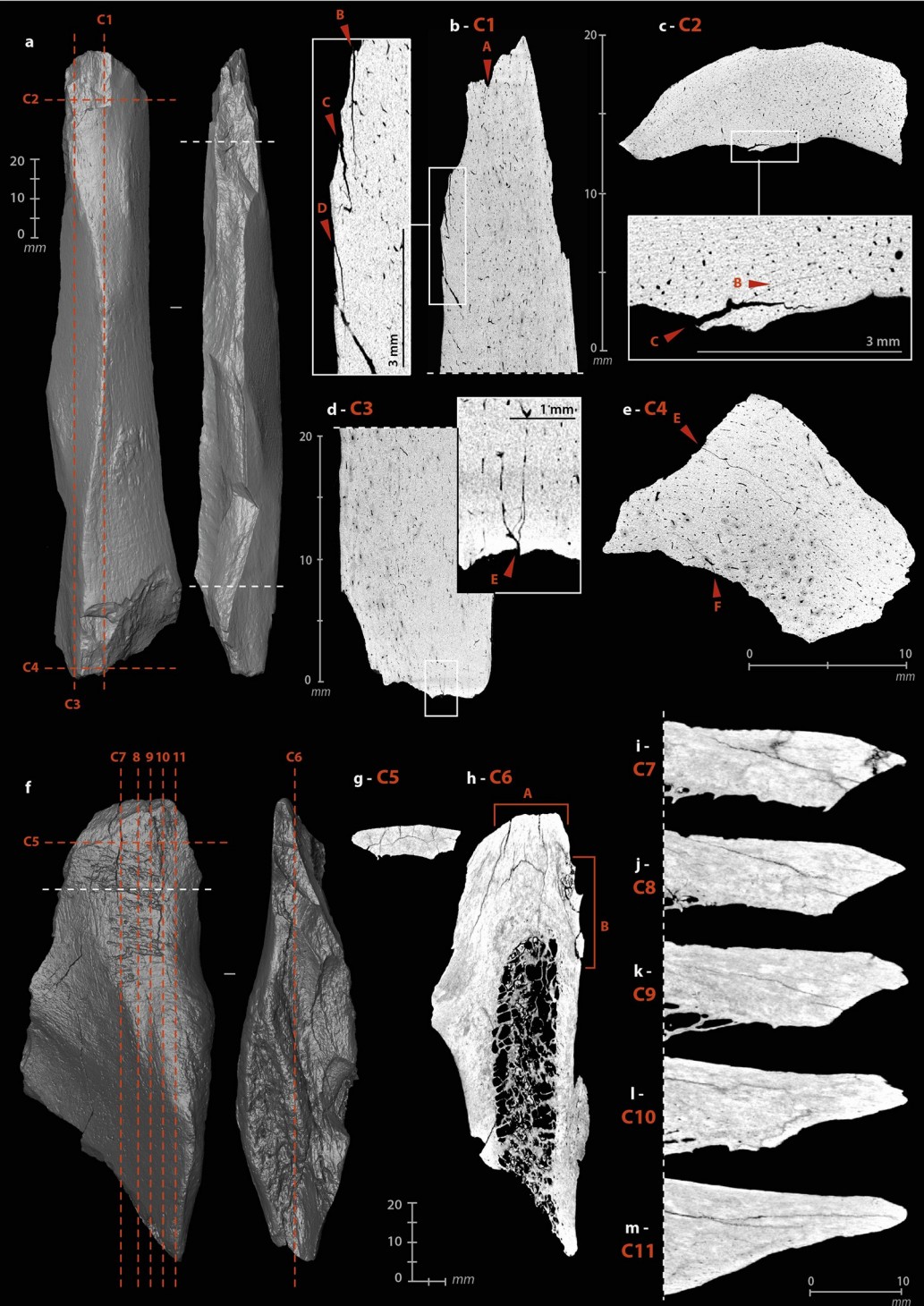

**Fig 9. µCT imaging of beveled tools.** (a) 3D reconstruction of an experimental sample from a Cow tibia. (b) cracks from use under the bevel, on a sagittal cross-section. (c) cracks from use under the bevel, on a transverse cross-section. (d) cracks from use under the striking surface, on a sagittal cross-section. (e) cracks from use under the striking surface, on a transverse cross-section. (f) 3D reconstruction of an archaeological sample from horse humerus, n˚1014. (g–h) concentration of cracks under the bevel, on a frontal cross-section. (i–m) sagittal cross-sections of the bevel showing the non-uniform damage distribution (µCT IP and CAD: M. Baumann).

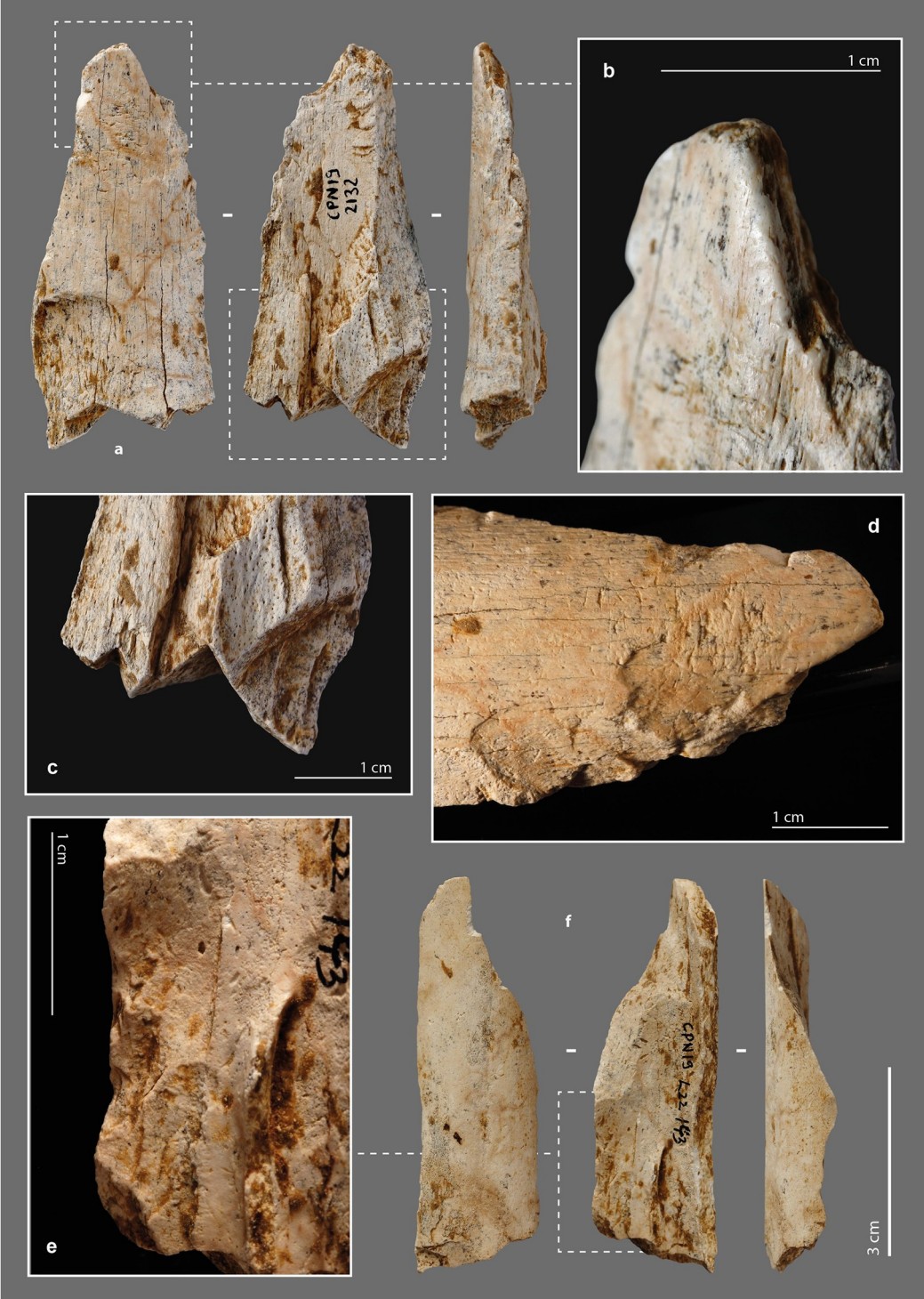

**Fig 10. Retouched tools from the Chez-Pinaud site, 2019–2020 excavations.** (a) possible beveled tool on large size ungulate diaphysis, n˚2132. (b) blunt localized at the distal end. (c) secondary fracturing step of the proximal end. (d) removals on the cortical surface at the distal end. (e) removals at the proximal end on the medullary surface. (f) possible beveled tool on reindeer metacarpal, n˚143 (photo: M. Baumann).

two on the cortical surface (Fig 10d) and one on the medullary surface, are large and scaled. The second series of 4 to 6 removals consist on an alternating and abrupt retouch, which formed a serrated profile edge. The latter, partially blunted, could correspond to an active edge. It should be noted that the use of the tool as a retoucher is posterior to its shaping. On the second specimen (Fig 10f), the retouch is located on the medullary surface of the proximal part (Fig 10e). It is a continuous retouch of at least three series of removals that form a straight edge. The opposite edge, also interrupted by a post-depositional break, has two low-angle removals, suggesting similar processing for both edges. We have not identified any use-wear traces. Thus, one of the hypotheses to put forward could be that of an adjustment for handling.

There are two small fragments of retouched bone artifacts (Fig 11). On the first one, the cortical surface and a fracture (occurring on fresh bone) form an acute angle where the retouch was performed. The latter is prior to the separation of this small fragment from a larger one. The retouch is scaly, low-angle, bifacial on a first portion and then unifacial (Fig 11b). The bifacial retouch is separated from the unifacial retouch by an unretouched but regularly blunted edge portion. The rectilinear leveling of the edge, its asymmetrical blunting, and the extension of the latter in the retouched areas suggest that it resulted from a transverse motion of the tool (i.e., perpendicular to the edge) in a positive rak angle, probably on soft material. This motion was still applied after the tool was retouched (Fig 11c). The second small fragment is a flake coming from a larger retouched tool. Its obverse side is the medullary face, its reverse is the flaked surface associated with part of a fracture from the first stage of fracturing of the blank. The retouch was done on the medullary surface (Fig 11d). It consists of several low-angle removals that form a sharp active edge. This active edge is slightly blunted with adjacent "stretched" and parallel reliefs, longitudinal striations and crescent-shaped chips (Fig 11f). It was probably used to cut middle soft material. The two small fragments thus belonged to larger tools whose active edges were shaped and/or resharpened by retouch.

Two nearly complete artefacts show removals on a lateral edge, part of which predates the adjacent fracture. This pattern involves the following sequence: (1) an initial fracturing aimed at obtaining the blank, (2) followed by the retouch of an edge, (3) a fracturing step that removed part of the retouched edge, (4) and then an extension of the retouch to the remaining part of the edge. This could correspond to a knapping accident. On the fragment also used as a retoucher and beveled tool (Fig 7a), 7 partially overlapping removals on the cortical face are present. The removals are initially invasive and semi-abrupt. Where the striking platform is preserved, they are scaled at low-angles (Fig 7b). The organization of the removals of a second fragment (Fig 2n) is comparable. On the portions of the edges where the striking platform is preserved, there is no evidence of use. The obtuse angles of the retouched edges and, in one case, the proximity to cancellous bone, do not make them suitable for use. The retouch would be better suited to shaping for sizing and/or handling.

**Internal damage.** The retouch, achieved by percussion, can cause cracks and micro-cracks in the bone. However, the pattern of cracks should be expected to be significantly different from other artifacts. The stress is not applied on a surface and in the direction of the mass (likely to absorb a significant portion of the energy), as with the retouchers, but tangentially along an edge. µCT recordings were made on five archaeological tools (S1, S2, S4, S5 and S7 Files) and two experimental tools (S2 Table).

In the experimental tools, cracks and micro-cracks developed from the percussion point. They are not systematic and vary in size from one to several millimeters (Fig 12b and 12c). They are always wider in the longitudinal axis than in the transverse axis and have a curved trajectory. The areas with the most numerous and extensive cracks correspond to those parts at the edge where most of the removals were knapped. Single cracks sometimes appear at the end of the removal in direct continuation of the flaked surface (Fig 12d).

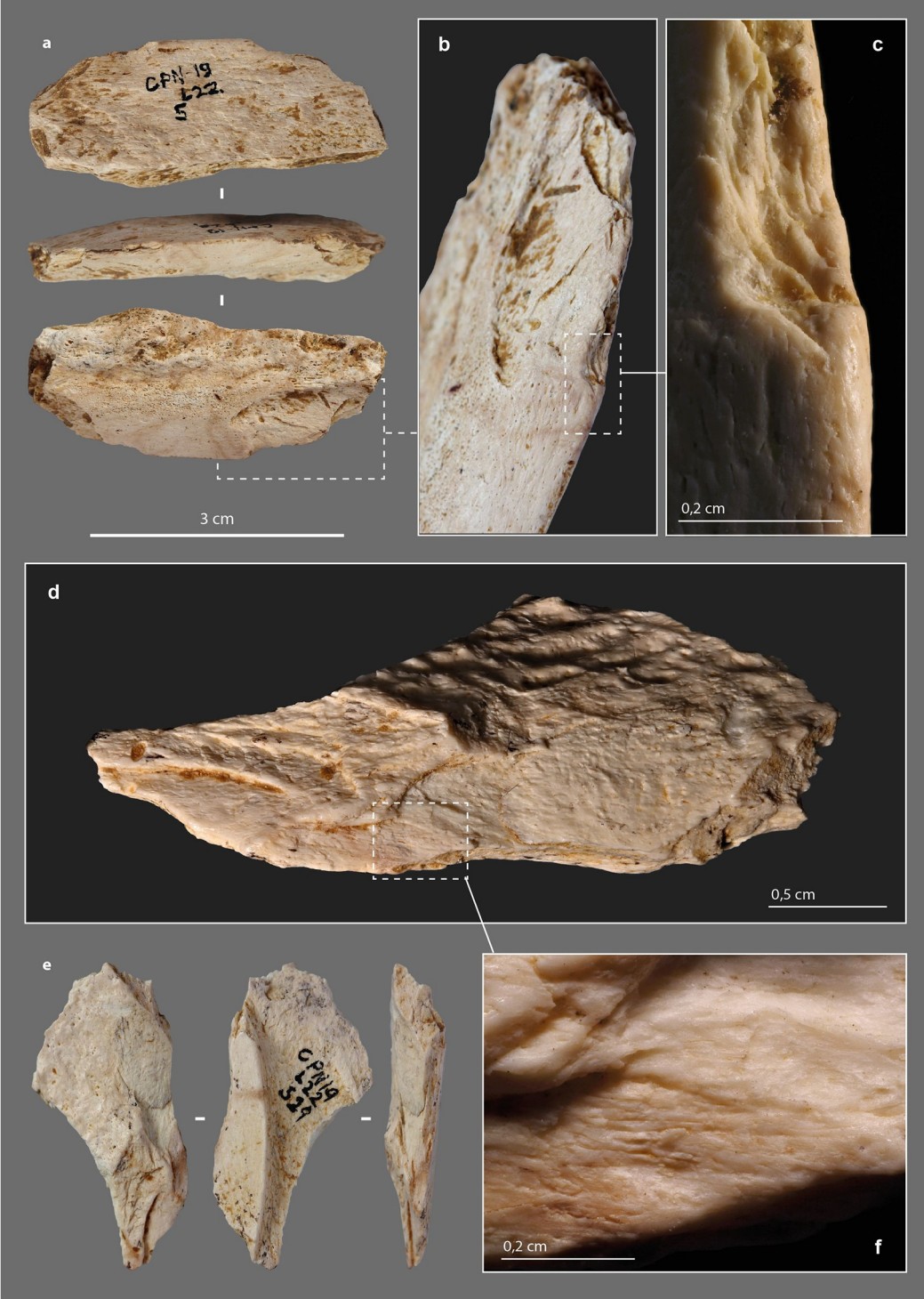

**Fig 11. Retouched tools from the Chez-Pinaud site, 2019–2020 excavations.** (a) retouched edge fragment on large size ungulate diaphysis, n˚5. (b) scalariform and low-angle removals. (c) retouch intercalated between 2 blunting episodes. (d) low-angle removals on the medullary surface. (e) retouched edge fragment on medium size ungulate diaphysis, n˚529. (f) detail of the blunt with longitudinal striations (photos: M. Baumann except d and f, H. Plisson).

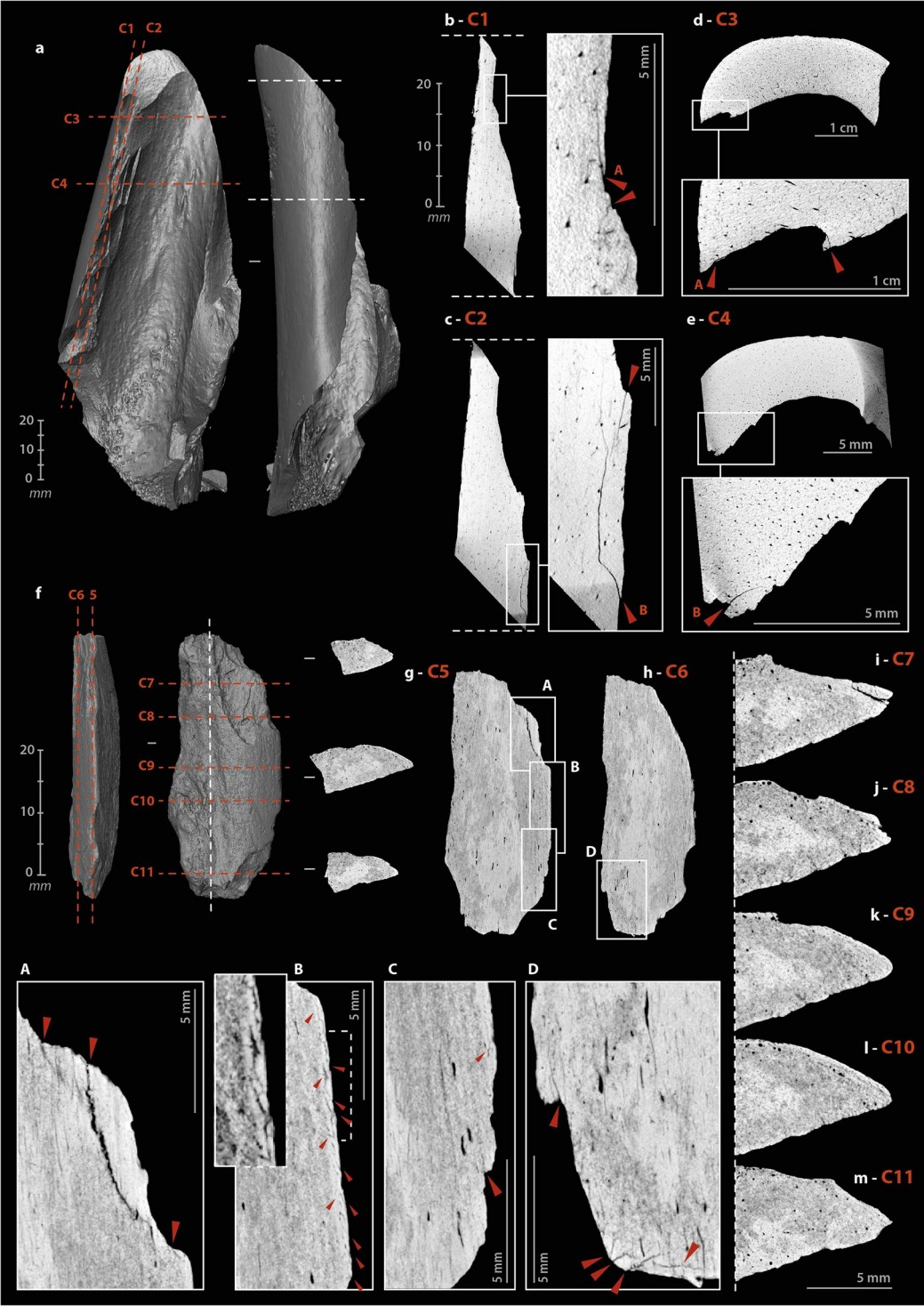

**Fig 12. µCT imaging of retouched tools.** (a) 3D reconstruction of an experimental sample from a Cow tibia. (b) multi-cracks from shaping at the percussion point on a sagittal cross-section. (c) simple cracks from shaping at the percussion point on a sagittal cross-section. (d) cracks from shaping at the beginning and the end of a removal. (e) simple cracks from shaping at the percussion point on a transverse cross-section. (f) 3D reconstruction of an archaeological sample from large size ungulate diaphysis, n˚5. (g-h) frontal cross-sections of the archaeological sample n˚5, A and C, cracks from shaping at the beginning of retouches, B, set of micro-cracks from use, D, multiple "star-shaped" cracks. (i–m) variation of the active edge morphology on transverse cross-sections (µCT IP and CAD: M. Baumann).

On archaeological specimens, cracks also developed at percussion points (Fig 12g, A), rarely at the end of removals. The micro-damage is more numerous and diverse than on the experimental specimens, probably due to a more complex history, including the use of the retouched edge, its resharpening, possible accidental breaks, taphonomic alterations, etc. One tool (Fig 12f) shows micro-cracks that are not related to the retouch but to the use of the edge. On the unretouched part of its active edge, a series of micro-cracks has developed directly below the surface, over a thickness of less than one millimeter with the same pattern. Here, the density of material, visualized on the μCT cross-section by a clear increase in grey levels, is higher than in the rest of the tool (Fig 12g, B). If this pattern correlates with surface damage, it could correspond to transverse motion of the bone tool on soft material in a positive rak angle. Transverse μCT cross-sections of the edge (Fig 12i–12m) confirm the pronounced and asymmetric nature of the blunt with the formation of a flat on the cortical face (Fig 12k). The retouch was probably done to restore some sharpness to the edge (Fig 12m). Internal cracks are also visible on the opposite edge, adjacent to the fracture that probably caused the fragment to detach from its original tool. The multiple "star-shaped" crack could be related to a percussion point (Fig 12h, D).

## Smooth-ended tool

**Morphometry.** One of the faunal remains under study shows at its distal end a blunt that is characteristic of use (S4 Text, S8 File). It is a portion of the mesial part of a large ungulate rib (L. = 9.44 cm; W. = 1.68 cm; cortical Th. = 0,3 cm). The tool, rather brittle, was collected in several fragments that were glued back together. Three quarters of the lower face is missing, while the proximal extremity is a recent break (Fig 13).

**External modifications.** Intense scraping is visible on the dorsal, ventral, and cranial faces of the rib fragment (Fig 13c). At one extremity, a blunt covers a bending fracture that reveals the cancellous tissue (Fig 13a). At this location, the junction between the ventral surface and cancellous tissue is a flat surface (Fig 14c), whereas the junction between the dorsal surface and cancellous tissue is smoother and shows micro-flakes and tears (Fig 14d). At the top, the rounded compact surface is marked by indentations and micro-flaks (Fig 14a and 14b). Over the entire surface, the micro-flacks and indentations are partially covered by the blunt area. The latter is associated, on the dorsal and ventral faces, with striations, shorter than the scraping striations but always more or less parallel to the main axis (Fig 13b).

Experimentally, the sharpness of a bone edge decreases quite rapidly during its use, but its loss of matter is minimal unless the material being worked is abrasive. Macroscopic blunting of a sharp irregular bone edge cannot result from scraping soft organic material (e.g., skin, leather, bark). The latter would be ripped or pierced before a blunt had developed. The bluntness of the bending fracture and the nearly spherical shape of the extremity of the rib examined (Fig 14f), which is not in the progressive extension of the rib sides, indicate shaping by abrasion. The chronological relationship between this shaping, the scraping of the sides and the use-wear traces is unclear. The indentations and micro-flakes leveling may precede the final shaping stage, due to the semi-secant character of the various surfaces of the distal extremity. Perhaps are we dealing with the rejuvenation of a tip previously deformed by use? This combination of features is close to those of pressure flakers [158–162].

**Internal damage.** The active part of this rib segment results from a combination of shaping and use-wear. Abrasion shaping induces compressive stress, first applied randomly to the surface roughness. Once the roughness is reduced, the stress is distributed more evenly over a larger contact area which reduces the pressure. This shaping does not *a priori* induce internal cracks and micro-cracks. Conversely, indentations and micro-flakes are the consequence of a

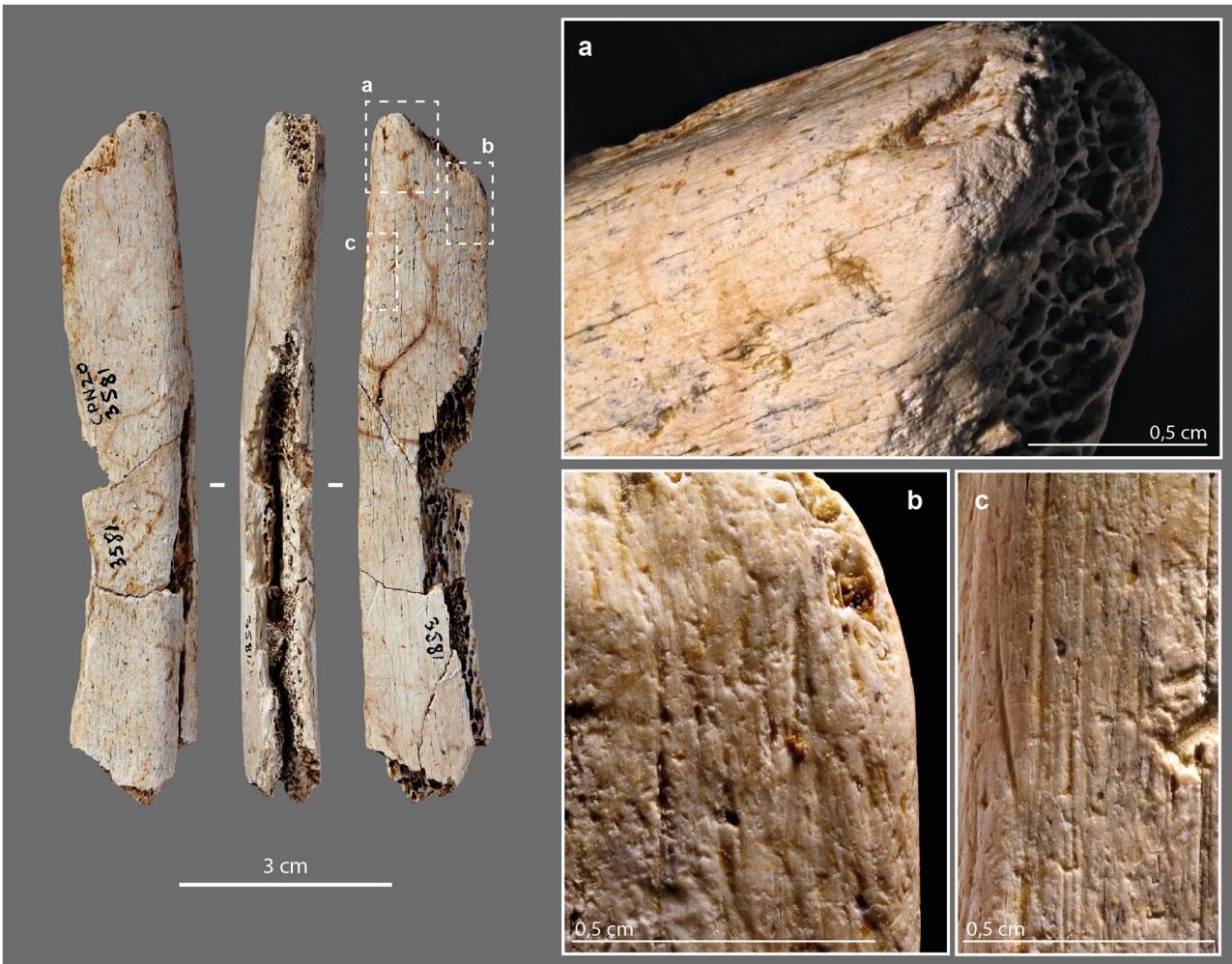

**Fig 13. Smoothed-end tool, n°2020, from the Chez-Pinaud site, 2019–2020 excavations.** (a) blunted bending fracture at the distal extremity. (b) striations from use on the ventral face at the distal extremity. (c) scraping striations on the cranial face (photos: M. Baumann).

point stress that has exceeded the elastic deformation capacity of the bone and is likely to produced internal damage. On the μCT cross-sections, the rib shows several transverse and longitudinal cracks. In its mesial and proximal parts, the through-cracks follow the same orientation as those that caused the post-depositional fragmentation. They always develop perpendicular to the cortical surface toward the cancellous tissue. At the active part, the only crack detected originates at one edge and traverses the cortical tissue toward the opposite edge and not toward the cancellous tissue (Fig 14g). This could be due to a different formation dynamic. In addition, this crack starts from an indentation (Fig 14e) and follows the same orientation as the micro-flake on the opposite side (Fig 14f). It could therefore result from the same type of stress as the latter.

## Discussion

### Towards an outline of the Neanderthal bone industry

Our main study objective was to identify, in the western side of the Neanderthal expansion zone, evidence of a common use of bone as a raw material for tool making, as we did in the

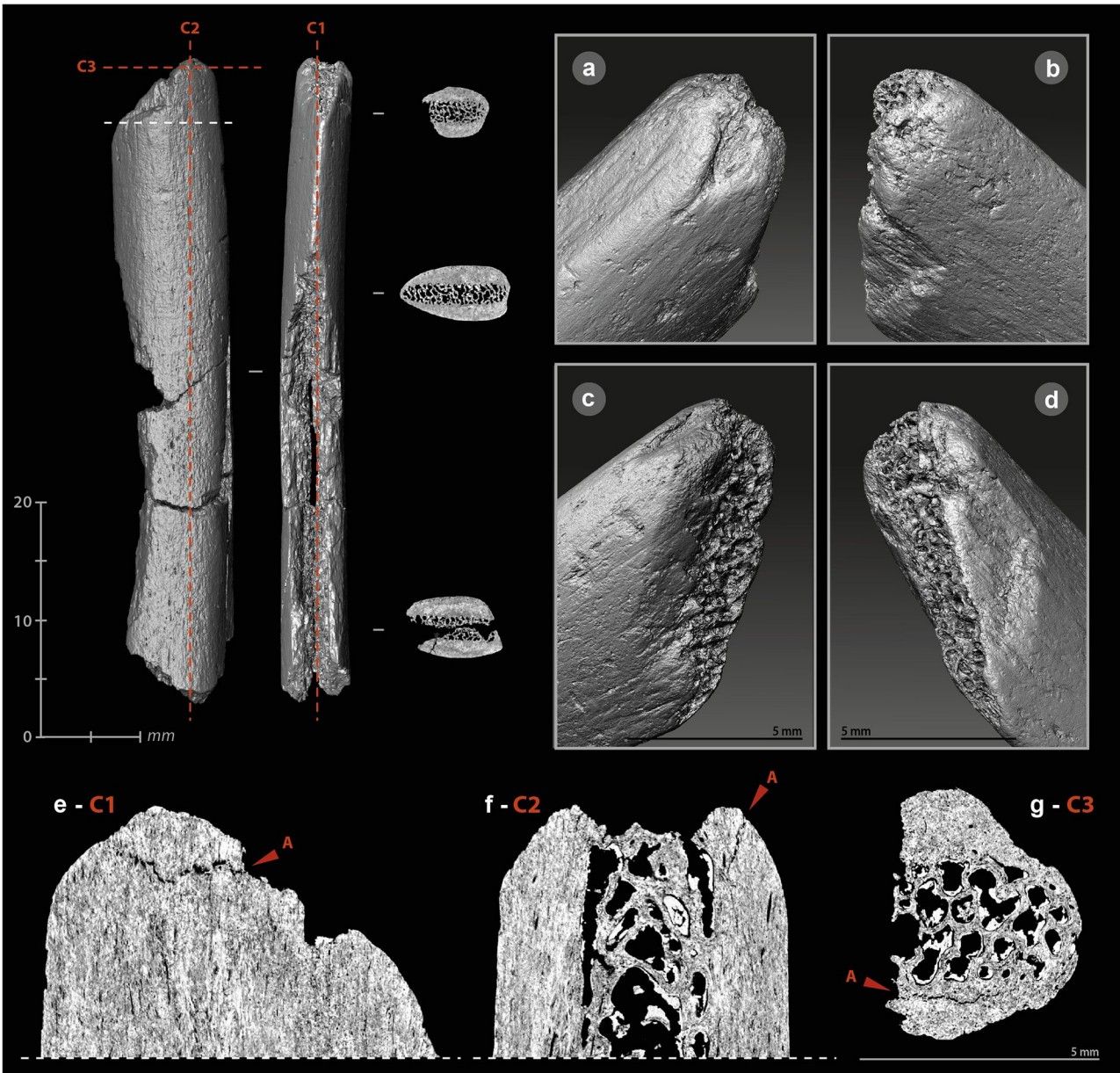

**Fig 14. µCT imaging of a smoothed-end tool, n˚2020, from the Chez-Pinaud site, 2019–2020 excavations.** (a–b) rounded and compacted surface marked by indentations and micro-flakes at the distal end. (c) flat surface at the junction between the ventral surface and the cancellous tissue. (d) micro-flakes and pull-outs on a smoothed surface at the junction between the dorsal surface and the cancellous tissue. (e–g) frontal, sagittal and transvers cross-sections showing a possible crack of use.

easternmost part in the Chagyrskaya Cave (Siberia, Russia). The identification of 103 bone tools among the 3220 faunal remains from the first two new campaigns at the Chez-Pinaud site provides such evidence. This number of tools is quite large given the small area currently excavated (7m$^2$), but more significantly, it is equivalent to the number of lithic tools (N = 109). Of the 103 bone tools, 83 are bone retouchers, similar to those previously identified at this site and comparable to those found in many Mousterian assemblages across Eurasia. The originality lies within the 20 additional tools used for other purposes: as wedge/chisel, lateral cutting

edges and, perhaps, pressure flaker. The Quina groups that occupied the Chez-Pinaud site thus produced a variety of tools for different tasks. Although bone tools are much more numerous at Chagyrskaya Cave [7], coming from a larger excavated area, the distribution of different categories is more or less the same, with a large majority of retouchers (88% at Chez-Pinaud; 94% at Chagyrskaya) followed by tools with lateral retouch (6.3% and 9.7%), beveled tools (4.7% and 6.8%), and smooth-ended tools (1.8% and 2%), with respectively 8.7% and 6.8% multi-use tools. These four types, although disparately, have been already reported, from other Mousterian sites, such as Axlor ("retocadores," "cincel," "hueso retocado," "alisador" [36]) and Combe-Grenal ("retouchoirs", "outils façonnés par percussion," "pièces esquillées," "pièces à extrémités émoussées" [28]).

It is not yet possible at this early stage of the study to determine whether the tool blanks are the result of integrated debitage within the food fracturing, a distinct operative sequence, or an opportunistic later collection. A study of the variability of bone processing (by species and anatomical elements) and an assessment of the degree of control over the morphometry of the blanks remains to be carried out. However, obtaining proper edge angles without controlled fracturing seems unlikely, leaving the question open. A trend is apparent in the characteristics of the blanks used. Long bones from large size ungulates (horse, bison) were mainly selected and a strong selection was made on the remains of medium size ungulates (reindeer), showing a clear preference for the largest size and/or the densest fragments. This preference is also very clear at the Chagyrskaya Cave, where more than 80% of the bone blanks are from large size ungulates, whereas they account for only 52% of the ungulate remains (NISP) from the Micoquian deposits [163]. More broadly, this pattern is consistent with the selection observed in Mousterian assemblages for the retouchers [138,144,164–168].

The shaping of the tools is clearly attested. It is mainly concentrated on the active parts. Scraping marks are visible on the active area of about 23% of the bone retouchers, a proportion close to those of the bone retouchers from the Quina deposits of De Nadale (17%) [168] and Les Pradelles (18%) [144]. Scraping was used to shape and/or clean the smooth-ended rib, while abrasion was used to shape and/or rejuvenate the tip. Only one specimen from the Chez-Pinaud site is currently available, but other smooth-ended ribs shaped by scraping and/or abrasion have already been reported in Middle Paleolithic assemblages dated between 75 and 45 ka ago, including those from Abri Peyrony and Pech de l'Azé [30], La Quina [41], Axlor [36], Zaskalnaya VI [21] (S3 Table, S3 Fig) or Lartet [169]. Within the Quina levels of Chez-Pinaud, direct percussion is widely applied, for shaping or resharpening the lateral edges and bevels and, possibly, for shaping out the prehensile parts. Among ancient bone artifacts, the most frequently reported are those that were retouched, although only the least questionable specimens, deeply transformed, are illustrated, such as the elephant bones bifaces [170] (S4 Table, S4 Fig). Following Anne Vincent's conclusions on the Charente Mousterian deposits [32], we note, at Chez-Pinaud and Chagyrskya, the tendency to limit the retouch to only a few removals, necessary and sufficient to make the active edge effective. A beveled artefact raises the question of the production of bone bladelets from core-like blanks. The evidence is tenuous but consistent with some of the bone artefacts from Chagyrskaya: burin-like tools (Fig 15d), possible bone core (Fig 15e) and bone bladelets (Fig 15c).

## A functional diversity to be explored

Bone retouchers form a homogeneous functional category, i.e., light hammers for retouching lithic edges. In US22, bone retouchers appear to be primarily dedicated to resharpening side scrapers during carcass processing. Waste from shaping and resharpening scrapers was primarily done with a soft hammer [65], and, among the removals, those related to maintenance

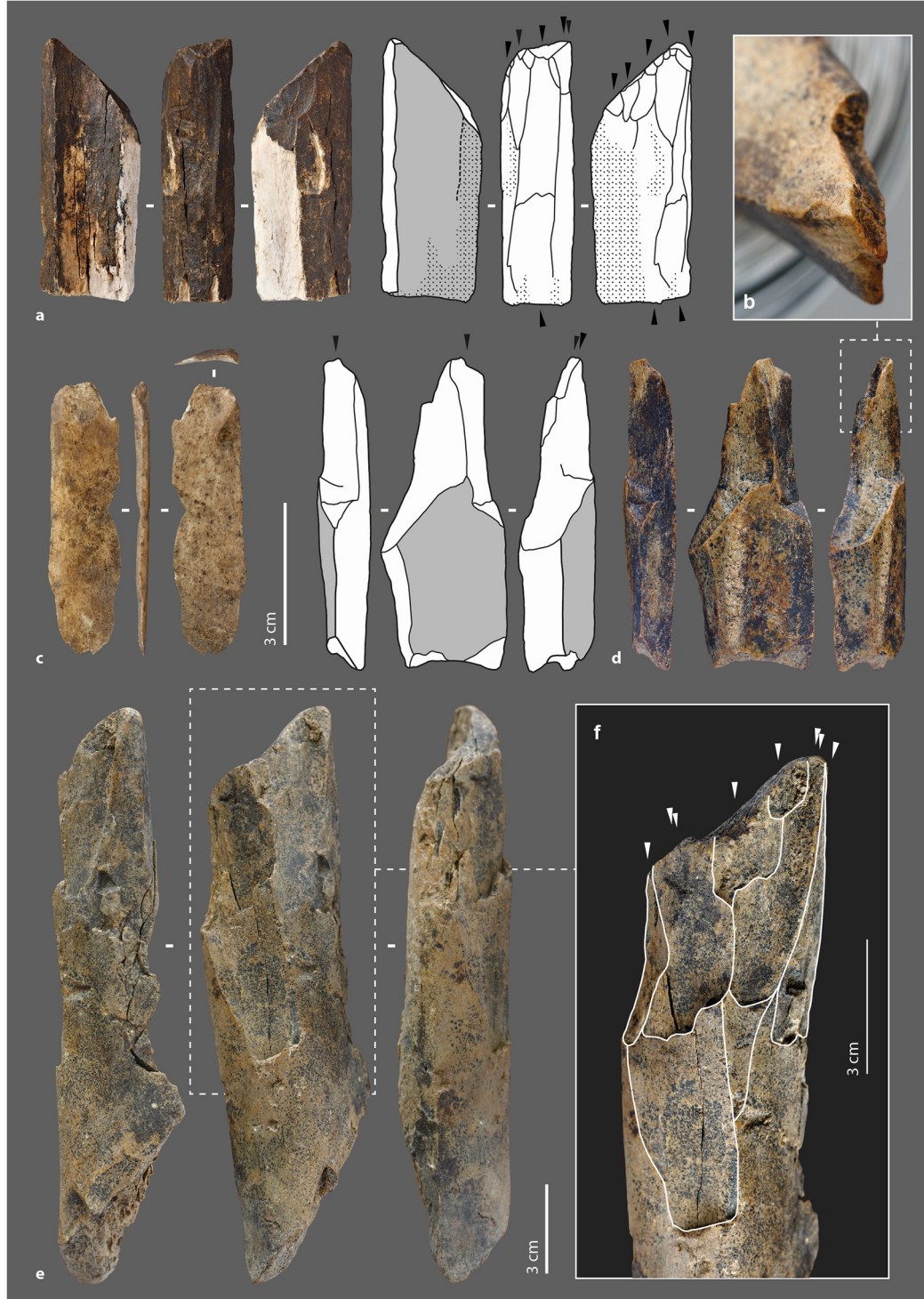

**Fig 15. Micoquian bone industry from the Chagyrskaya cave, Russian federation, 2008–2018 excavations.** (a) bone tool with negatives of lamellar removals. (b) detail of a burin-like removal negative. (c) bone bladelet. (d) burin-like tool. (e) possible bone nuclei. (f) negatives of lamellar removals (photos: M. Baumann).

dominate, while traceological analyses shows that the side scrapers were used to cut meat and hide [171], but a more precise correlation could be drawn between bone retouchers groups and specific knapping tasks [138,144,148]. For this purpose, the information provided by external damage and tools morphometry, on which Costamagno et al.'s bone retouchers groups are based [144], could be supplemented by internal damage detected by μCT. In particular, the cracks that developed under the scores areas can provide gesture information that would identify, for example, the bone retouchers used for the Quina retouch [149].

Among the beveled tools, the diversity in active edges (dimensions, position, front delineation etc.) and the distribution of use-wear traces, suggest that these tools were used in a significant diversity of tasks, involving different worked materials: soft, when blunted and without chippings, harder, when chipped with limited bluntness. The short length of complete specimens, the frequency of short and transverse shaft fractures, and the modification of the handling part of one of the tools could argue for the use of handles [110], as already demonstrated for Middle Paleolithic lithic tools [172] Handles, which improve the efficiency (accuracy and force) [173], are generally regarded as a significant investment of time and energy [174], as well as an essential step in the technological evolution [175,176]. However, a wooden clamp and leather strap as we have done experimentally is not a great investment.

Based on its mechanical properties, the bone responds favorably to experimental knapping. Edge shaping by percussion does not necessarily mean the placement of an active part. This may explain the heterogeneity in retouched tools, both in terms of morphometry and retouch quality. In all cases, the retouch was performed on thick blanks and the removals, preferentially on the medullary face, seem to be limited to what is strictly necessary. Two active edge fragments provide functional clues. Both were used on soft material, one for cutting and the other for scraping. In the latter, we observe that the pattern of internal damage can indeed change as a function of the stress experienced. Technical advantages of bone cutting edges, at a site where flint is abundant, remains to be understood.

Smooth-ended tools, particularly rib tool, are often considered as hide working [29,30,36,154,177], due to the regular rounded shape of the active extremities that can be interpreted as resulting from significant contact with a soft worked material. In our study, the Chez-Pinaud rib shows external and internal damage from a point contact area with a material harder than bone, with the regular rounded active edge being the result of shaping. The Chagyrskaya deposits have yielded, at least, two tools with similar characteristics [7]. Experimentally, many tasks can lead to the blunting [178]. It should be kept in mind that such task diversity could exist in the Middle Paleolithic.

## Understanding the role of bone tools in technical systems

The recurrence of bone tool types with distinct potential functionalities in the US22 from Chez-Pinaud, together with that of the Micoquian assemblage from Chagyrskaya and the occasional report of similar tools from numerous Middle Paleolithic sites, suggests that these elements had a specific role to play in the pre-Upper Paleolithic technical systems and that their apparent simplicity is not synonymous with opportunism. The equivalent proportions of bone and lithic tools in a task specific location, such as Chez-Pinaud, allows us to consider that the two productions had a technical synchrony. While the complementarity between the bone retouchers and the sharp lithic edges is fairly obvious in a site dedicated to carcass processing, the other functional categories of bone tools reflect a broader range of related activities that may have taken place at the site. The functional spectrum of the Quina flint industry is limited, as it relies on a low diversity of tool types, mostly short, often thick, lacking the pointed shapes of the biface or Levallois industries. Bone provides elongated shapes, sometimes with minimal

preparation (ribs), while its resistance to impact and relative elasticity make it better suited to certain tasks than flint. Such complementarity is not specific to Middle Paleolithic, but it appears all the more necessary in a technical context such as that of the Quina Mousterian.

Whatever the system considered, its functioning necessarily requires a certain stability despite its internal constraints (or contradictions) [179], which are predictable, and external which are fluctuating. Having a low-cost solution for the execution of an essential task meets this stability requirement. Possibly this is what using an easily renewable by-product of a central recurring resource provides. Nevertheless, at Chez-Pinaud the selection of diaphysis and ribs from the large size ungulates, as well as the larger modules from medium size ungulates, suggests that the raw material for tools making was not so abundant to cover all needs that the production of blanks could depend on random fracturing. For skilled flint knappers accustomed also to breaking bone, obtaining bone fragments of predefined shape and edge angulation was certainly not a concern. A diversity in Mousterian bone breakage pattern has been recently evidenced [180] which may not be linked with marrow recovering only. This is a technological topic to be investigated in itself, as M. Mozota Holgueras advised for retouchers [181].

Nearly 20% of the pieces show evidence of re-use, with use as a retoucher being the most recent. This opens up the notion of blank management. Although robust, the thickness of the horse and bison bone blanks, not to mention reindeer, did not allow for successive and alternative forms as observed for lithic tools circulating from site to site [65]. This intrinsic limitation, coupled with the recurrent availability of bones, means that the bone tools discussed here were likely produced, used and discarded in situ, as has been described elsewhere for certain categories of lithic tools [182].

## Conclusion

From the Altai to the Atlantic shore, through a multitude of sites where only a few objects have been reported so far, evidence of a Neanderthal bone industry is emerging, probably with ancient roots [183,184]. Its invisibility, even in the most recent synthesis published on Neanderthal [185], is probably due in part to ill-defined categorizations in which aesthetic and ideological anthropocentric criteria inherited from the last century are still combined. It is also related to a compartmentalization between the fields of expertise necessary for its identification: at Chez-Pinaud site, bone tools other than the retouchers are only highlighted in the part of US22 currently studied. Nevertheless, regardless of the cognitive value attributed to the use of bone material and its transformation techniques [169], bone tools shed light on related technical registers that are little or not documented through lithic tools and provides complementary information on Middle Paleolithic subsistence strategies. There is thus a promising avenue of research to be further explored.

## Supporting information

**S1 Fig. Archaeological deposits of the Chez-Pinaud site (Jonzac, France).** a) location of the excavation area in 2019–2021; b) left stratigraphic cut (after Airvaux and Soressi 2005); c) view of the "bone-bed", 2021 excavation (photos: W. Rendu).
(PDF)

**S2 Fig. Experiments with replicas of Mousterian type bone tools.** a) fracturation by direct percussion on anvil; b) fractured fresh long bones of Bovinae; c) regular flakes from bone fracturing; d) impact of direct percussion; e) retouched bone blank; f) flake from retouch; g) Mousterian side scrapers shaping; h) meat cutting; i) hair removing from skin; j) wooden

peeling; k) plant harvesting; l) soil digging (photos: H. Plisson).
(PDF)

**S3 Fig. Examples of ribs with smoothed end discovered in other Middle Paleolithic contexts.** a, c–d) Abri Peyrony; b) Pech de l'Azé (after Soressi et al. 2013); e) Zaskalnaya VI (after Stepanchuk et al. 2017); f) Chagyrskaya Cave (after Baumann et al. 2020); g) Abri des Canalettes (after Patou-Mathis 1993); h) Axlor (after Mozota Holgueras 2012); i–k) La Quina (after Henri-Martin 1907–1910); l–m) Grotte du Noisetier (after Oulad El kaïd 2016).
(PDF)

**S4 Fig. Examples of retouched bone artifacts discovered in pre-AMH contexts.** a) Castel di Guido (after Villa et al. 2021); b) Nova de Columbeira (after Zilhão et al. 2011); c) Vauffrey (after Vincent 1993); d) Poggeti Vecchi (after Aranguren et al. 2019); e) Bois-Roche (after Vincent 1993); f) Combe-Grenal (after Tartar and Costamagno 2016); g–h) Chagyrskaya (after Baumann et al. 2020); i) Gran Dolina (after Rossel et al. 2011); j) Abric Romaní (after Carbonel et al. 1994).
(PDF)

**S1 Table. Microtomographic recording parameters of the experimental and archaeological tools from Chez-Pinaud site.**
(PDF)

**S2 Table. Experimental samples analyzed in μCT.**
(PDF)

**S3 Table. Reports (non-exhaustive) of smoothed-end ribs from Neanderthal contexts in Eurasia.**
(PDF)

**S4 Table. Reports (non-exhaustive) of knapped bone tools from pre-AMH contexts in Eurasia.**
(PDF)

**S1 File. 3D model, retouched bone tool #CPN19-5, Chez-Pinaud site.**
(PDF)

**S2 File. 3D model, retouched bone tool #CPN19-529, Chez-Pinaud site.**
(PDF)

**S3 File. 3D model, beveled bone tool #CPN19-534, Chez-Pinaud site.**
(PDF)

**S4 File. 3D model, beveled bone tool #CPN19-888, Chez-Pinaud site.**
(PDF)

**S5 File. 3D model, beveled bone tool #CPN19-1014, Chez-Pinaud site.**
(PDF)

**S6 File. 3D model, bone retoucher #CPN19-2020, Chez-Pinaud site.**
(PDF)

**S7 File. 3D model, retouched bone tool #CPN19-2132, Chez-Pinaud site.**
(PDF)

**S8 File. 3D model, smoothed-ended bone tool #CPN20-3581, Chez-Pinaud site.**
(PDF)

**S9 File. 3D model, beveled bone tool #CPN20-3609, Chez-Pinaud site.**
(PDF)

**S1 Text. Bone retouchers.**
(PDF)

**S2 Text. Beveled tools.**
(PDF)

**S3 Text. Retouched tools.**
(PDF)

**S4 Text. Smooth-ended tools.**
(PDF)

## Acknowledgments

The Chez-Pinaud excavation is under the supervision of the Service Régional d'Archéologie of Nouvelle Aquitaine, Poitiers. We are grateful to the Conseil Général of the Charente-Maritime for its support. The field work will not be possible without the major help of the Communauté de Commune de Haute Saintonge and Jonzac Municipality. We thank the two reviewers whose comments contributed to the improvement of this article.

## Author Contributions

**Conceptualization:** Malvina Baumann, Hugues Plisson.

**Data curation:** Nicolas Vanderesse, Guillaume Guérin, William Rendu.

**Formal analysis:** Sylvain Renou, Nicolas Vanderesse, William Rendu.

**Funding acquisition:** Malvina Baumann, Hélène Coqueugniot, Ksenyia Kolobova, Guillaume Guérin, William Rendu.

**Investigation:** Malvina Baumann, Hugues Plisson, Serge Maury, Sylvain Renou, Nicolas Vanderesse, Ksenyia Kolobova, Svetlana Shnaider, William Rendu.

**Methodology:** Malvina Baumann, Hugues Plisson, Serge Maury, Nicolas Vanderesse.

**Resources:** Serge Maury, Hélène Coqueugniot, Nicolas Vanderesse, Veerle Rots.

**Supervision:** Hugues Plisson, Hélène Coqueugniot, Veerle Rots, William Rendu.

**Validation:** Hugues Plisson, Hélène Coqueugniot, Veerle Rots, William Rendu.

**Visualization:** Malvina Baumann.

**Writing – original draft:** Malvina Baumann, Hugues Plisson.

**Writing – review & editing:** Serge Maury, Sylvain Renou, Hélène Coqueugniot, Nicolas Vanderesse, Ksenyia Kolobova, Svetlana Shnaider, Veerle Rots, Guillaume Guérin, William Rendu.

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
