## [Decision Letter · Decision Letter 0]

19 Dec 2022

PONE-D-22-30527On the Quina side: A Neanderthal bone industry at Chez-Pinaud site, FrancePLOS ONE

Dear Dr. Baumann,

Thank you for submitting your manuscript to PLOS ONE. After careful consideration also based on the reports produced by the reviewers, we feel that it has merit but does not fully meet PLOS ONE’s publication criteria as it currently stands. Both reviewers consider this work worthy of publication although in the meantime they request that you tackle some issues in the structure of the article. These particularly affect the first part of the manuscript, where unnecessary data, unclear sentences and statements hamper the fluency of reading. Moreover, you are invited to clearly state which gaps currently concern the study of bone technologies in the Middle Palaeolithic and to avoid incertitudes about the criteria you used to isolate the bone fragments from the faunal assemblage, to previously identify them as tools and using them as materials for the whole study, given the lack of common criteria previously applied by different scholars. Reviewers also requested to better explain the experimental trials and re-arrange the discussion section. Therefore, we invite you to submit a revised version of the manuscript that addresses the points raised during the review process.

We look forward to receiving your revised manuscript.

Kind regards,

Marco Peresani

Academic Editor

PLOS ONE

Journal Requirements:

2. In your manuscript, please provide additional information regarding the specimens used in your study. Ensure that you have reported specimen numbers and complete repository information, including museum name and geographic location.

If permits were required, please include the following statement:

For more information on PLOS ONE's requirements for paleontology and archaeology research, see https://journals.plos.org/plosone/s/submission-guidelines#loc-paleontology-and-archaeology-research.

"The Chez-Pinaud excavation is funded by the Ministère de la Culture under the supervision of the Service Régional d’Archéologie of Nouvelle Aquitaine, Poitiers. We are grateful to the Conseil Général of the Charente-Maritime for its financial and logistic support. The field work will not be possible without the major help of the Communauté de Commune de Haute Saintonge and Jonzac Municipality. This research was supported by the Marie Sklodowska-Curie actions under the European Union’s Horizon 2020 research and innovation programme (Grant agreement #839528).  This article is part of the ERC QuinaWorld project (Starting Grant #851793). The Chagyrskaya Cave study have been supported by the Russian Science Foundation (Grant agreement #21-18-00376). We are thankful to the joint Russian-French Scientific Project of the Russian Foundation for Basic Research (Grant agreement #19-59-22007) and the Fondation Maison des sciences de l’Homme (FMSH) for financial support of the microtomographic analysis. These collaborations were initiated in the framework of the International Research Laboratory ARTEMIR. "

"Authors who did not recived a specific funding : Hugues Plisson; Serge Maury; Sylvain Renou; Hélène Coquegniot; Nicolas Vanderesse; Ksenyia Kolobova; Svetlana Shnaider; Veerle Roots; William Rendu

Funded studies : 

- M.B. was supported by the Marie Sklodowska-Curie actions under the European Union’s Horizon 2020 research and innovation programme (grant agreement #839528)

-GG was supported by the European Research Council (ERC) under the European Union’s Horizon 2020 research and innovation programme (ERC Grant agreement #851793)

The funders had no role in study design, data collection and analysis, decision to publish, or preparation of the manuscript"

Reviewers' comments:

Reviewer's Responses to Questions

**Comments to the Author**

1. Is the manuscript technically sound, and do the data support the conclusions?

Reviewer #1: Yes

Reviewer #2: Partly

2. Has the statistical analysis been performed appropriately and rigorously? 

Reviewer #1: Yes

Reviewer #2: N/A

3. Have the authors made all data underlying the findings in their manuscript fully available?

Reviewer #1: Yes

Reviewer #2: Yes

4. Is the manuscript presented in an intelligible fashion and written in standard English?

Reviewer #1: Yes

Reviewer #2: Yes

5. Review Comments to the Author

Reviewer #1: The article “On the Quina side: A Neanderthal bone industry at Chez-Pinaud site, France” by M. Baumann et al presents results of original research finalised to the identification of a Neanderthal bone industry in the Quina Middle Palaeolithic site of Chez-Pinaud. The study presents new results comparable with other contemporaneous sites, and the authors cite the existing literature accordingly. Data are presented in sufficient detail, especially raw data in the Supplementary Information section. The authors should be praised for the quality of the figures, which is truly remarkable.

However, I recognise some issues regarding the structure of the article: unnecessary data and unclear sentences, particularly in the first part of the manuscript, somehow shift the focus from the aims of the research. I would recommend the publication of this article only after a general polishing of the manuscript and the addition of more structured, clear statements regarding the aims of the research. Consequently, the Discussion section should be restructured accordingly.

Introduction

This is the most problematic part of the manuscript. Although the authors give an excellent framework of the gaps currently affecting the study of Middle Palaeolithic (MP) bone technologies, they do not state clearly enough what is their proposal to fill these gaps. Is the aim of this research to address the lack of ‘shared criteria’ in the study of MP bone industries by proposing a new analytical methodology (micro-CT)? More specifically, is it to give guidelines to other researchers to identify (in the words of the authors) a ‘genuine bone industry’? Or is it to replicate the methods previously proposed by the authors (Chagyrskaya) and to apply those methods to a Neanderthal context in Western Europe, since Chagyrskaya and Chez-Pinaud are both Quina sites? Some structure and rephrasing are necessary. I suggest, among other things, (1) giving a clear, technical definition of what the authors mean by ‘bone tool’ and ‘bone industry’, (2) giving a summary of methods and results applied in the study of the Chagyrskaya bone industry, and state why we should follow a similar approach for Chez-Pinaud, and (3) use a more dubitative tone throughout the Introduction, to communicate that previous works do exist and are valid, but need updating. The authors have done excellent work with the Supplementary Information, and most of what is missing from the manuscript can be taken from there.

Following, are some specific comments:

• Line 66: the sentence is unclear. Do you mean, it is the only bone industry surely made by Neanderthals? This is the part where I believe a clear definition of ‘bone tools’ would be appropriate. In lines 65-81 there is a list of previously identified bone tools ‘categories’, but it results as a mix of retouchers, lissoirs, scrapers, etc without ever addressing the general definition of ‘bone tool’. We all agree that Neanderthals can make a ‘proper’ bone industry -- but what is a ‘proper’ bone industry in fact? Because it is such a big question, citing previous research addressing this issue might not be enough in this case.

• Line 78: ‘most of them recently investigated’ might be more appropriate.

• Line 91: I would rather say that a proper study of MP bone tools (not bone industry) is anecdotal. It might be true that bone industries are only sporadically present in the MP -- we actually don’t know it because we never created a methodology to investigate bone tools! I believe the key to this sentence should be to critique the previous approaches to the study of bone tools, in fact anecdotal and most of the time borrowed from lithic studies.

• Line 92: ‘mode of manufacturing’ sounds confusing. Maybe ‘manufacturing techniques’? There is also a grammatical discrepancy (plural ‘their’ / singular ‘bone’).

• Line 94-96: give some literature references to other studies highlighting the issue of the lack of shared identification criteria (some are cited elsewhere in the Introduction).

• Line 98: briefly summarise the results from your previous study. Moreover, ‘evidence in the literature’ might be more appropriate than ‘bibliographic data’.

• Line 99: ‘genuine’ is somehow a foggy term, I suggest rephrasing it.

• Line 100: ‘primary use of percussion’ is also a little foggy. What do you mean by ‘primary’?

• Line 107-108: like in my previous comment (line 98) please openly state what the ‘Chagyrskaya analytical framework’ is.

• Line 108: something like ‘a site characterised by Neanderthal occupation’ would sound better to the reader.

• Line 111: here ‘bone tool’ should be replaced with ‘bone industry’.

• Line 116-118: it is unclear to me why we should compare Chagyrskaya and Chez-Pinaud. If you want to apply the same methods from Chagyrskaya, why did you choose Chez-Pinaud and not any other site with Neanderthal occupation and previously identify bone retouchers and bone-bed?

Methods

In general, I would appreciate more depth in the methods, since the main aim of the authors is supposed to be to encourage the creation of a shared methodology for the study of bone tools. Particularly, I am confused about the criteria the authors used to isolate the 103 bone fragments from the faunal assemblage by giving them the definition of ‘tools’ and using them as materials for the whole study. As we understand from the Introduction, we do not have any shared criteria to do so at the moment, and the authors need to share the criteria they used. Most of the requested information is in the SI, but I believe stronger statements are needed here in the manuscript. The same goes for details on the experimentally produced bone tools. I do not think the whole experimental protocol belongs to the manuscript, but perhaps some justification of why you need the experimental bone tools within your analysis might fit well in the Methods section.

• Line 138-140: unclear sentence. Does it mean that during the last excavation lithics were recovered and subjected to preliminary studies which revealed they belong to Quina industries?

• Line 147-148: are there any other indicators of the cold, dry and open environment, apart from reindeer remains? If so, please cite them. Moreover, I am confused if these data refer to US 22 only, or a general description of the occupation of the site.

• Line 158-159: ‘bones anatomical connection’. Unclear. Do you mean ‘bones refitting’ or ‘bones which are anatomically connected (articulated) in situ’?

• Line 182-183: it seems to me somehow circular logic here. If the aim of the study is to identify bone tools among faunal assemblage by using a specific methodology, how can we state, in the methods, that 103 bone tools were found? Maybe this sentence belongs to the Results section.

• Line 202: what are the criteria used to select these 9 bone tools for the micro-CT? Did you choose one tool per ‘bone category’? Is it because of better preservation? Specific spatial distribution in the site? Please be more specific.

Results

I am confused by the description of bone tools from recent and old excavation campaigns. Do the results concern only recent findings (US 22), or do they also re-examine bone tools previously identified in older campaigns, but with the new ‘methodological framework’ proposed by the authors? Some clarification is needed. If, in fact, they do re-analyse bone tools previously identified, the authors should take time in the discussion to assess the validity of their new proposed methodology against the ‘unshared criteria’ used until now in the study of MP bone technologies. If the results of the present work mainly focus on bone tools coming from recent excavation, then I suggest deleting any reference to tools collected in previous campaigns, to avoid confusion. Perhaps in the future, the authors might want to consider undertaking a comparative study with bone tools described in the past to determine how their method is an improvement on what people did before, and how we can build new, shared criteria for the study of bone tools based on the methods the authors propose. Moreover, I found myself confused while reading the results about use-wear and manufacturing traces descriptions: it is not always clear if the authors are talking about traces observed on archaeological bones, rather than experimental bones, rather than results obtained in previous studies. Perhaps a more structured organisation of this section might be helpful, and comparisons within traces and tools might find a more suitable space in the Discussion.

• Fig. 2: a more schematic organisation of the identified tool ‘types’ might help the reader to get a visual understanding of the differences between the ‘types’ themselves.

• I question the use of the word ‘type’ to identify a tool showing specific features on which we can base the tool’s function (retouchers, smooth-ended tools, etc.). To me, the term ‘type’ carries too many reminiscences from the lithic typological approach (Bordes sensu), which several scholars consider nowadays an obsolete approach even for lithic technology. More importantly, it could be argued that the use of this terminology is inconsistent with what the authors stated in the Introduction, blaming the lack of proper methods for the study of bone tools on reckless borrowing from pre-existing lithic studies. The same goes for ‘tool category’. I suggest the authors might reconsider terminology here; for instance, in other parts of the manuscript, the authors use the term ‘morpho-functional category’, which I believe is the most appropriate definition the authors used so far.

• Line 255: a picture of these percussion marks and notches, even only in SI, could be appropriate. It could contribute to the debate regarding the attribution of percussion marks and notches to butchering activities rather than the use of bone blanks as retouchers (see: Delphine Vettese, Camille Daujeard. Le mythe du retouchoir en os : analyse d'un reste actuel / The myth of the bone retoucher: analysis of an actualistic bone referential. Colloque en hommage à Emilie Campmas "Sociétés humaines et environnements dans la zone circumméditerranéenne du Pléistocène au début de l'Holocène", Mar 2021, Toulouse, France).

• Line 261-268: a comprehensive table of these metric data in SI might help future researchers to compare their bone industry with yours.

• Line 280-282: tibias indeed are the easiest to recognise, but they are also the easiest bone to fracture and the bone that usually produces the greatest number of blanks upon breakage. Perhaps your observations are not biased…

• Line 283 (and 289, 292, 316, 381…): I would move all the comparisons with other Quina sites in Discussion.

• Line 304-306: same -- I would move comparisons with retouched lithic tools in Discussion.

• Line 331-343: this section might belong to the Methods.

• Line 346: please give a definition of ‘simple and complex micro-cracks’ in the Methods.

• Line 364-375: explanatory parts of this section should be moved to the Methods, whereas actual data and results should be left here.

• Line 390-391: do you have any experimental evidence for this fracture to be resulting from the tool’s use?

• Line 412-419: this should be in the Discussion.

• Can the authors spend a few words explaining how we can distinguish the manufacturing marks associated with ‘retouched tools’ from the percussion marks associated with the bone breaking during butchering activities? Retouched bone blanks are widely present in Quina sites, and a reliable traceological identification protocol is needed.

• Line 533: can you elaborate on this hypothesis?

• Line 549: please reference previous studies or the authors’ experimental evidence for this use-wear in association with cutting soft materials.

• Line 601-602: ‘including the use of the retouched edge’. I would like to see an experimental test, followed by micro-CT, of the retouched edge after the use proposed by the authors. The discussion about internal damage due to use-wear and use of the retouched edge is fascinating and deserves to be verified.

• Line 620: ‘must’ might be a little strong…

• Line 649: this claim is a little too blunt and needs more justification. What use? Which bone? Which material is being worked?

Discussion

After working on the previous sections, I believe there will be several topics the authors will want to address in the Discussions. At the moment, this section is somehow meagre and disorganised. I suggest giving a proper structure to this section through thematic headings (e.g., comparisons with other Quina sites, the potential of the analysis of internal damage, multifunctionality/re-use of bone tools in Quina culture compared to the lithic industries, suggestions for the creation of ‘shared criteria’ to study bone tools…).

• Line 685: why is it important to state that the number of bone tools is equal to the number of lithic tools?

• Line 753-754: I would like to hear more details on this. Why not? What would be a better investment instead?

• Line 763-764: what is your hypothesis on this, based on your results, observations and comparisons with other sites? You somehow address it in lines 784-789, but a little of a ‘dreaming’ speculation here couldn’t hurt.

• Line 766-773: very interesting!

• Line 778: ‘pre-Upper Palaeolithic’ might be misleading. Please rephrase.

• Line 801: typo. Replace ‘narrow’ with ‘marrow’.

• Line 1255: correct spelling is ‘Martellotta’ :)

In conclusion, I believe this is much-needed work because we require a shared methodology in the study of MP ‘non-official’ bone tools, as well as a reliable methodology to assess the multifunctionality of these tools. Throughout the whole paper, I very much appreciated the focus on traceological descriptions and mechanical properties of osseous materials. The application of the micro-CT to traceological questions is a fascinating subject, and its brilliance could and should be more emphasised in the paper. The authors’ proposal to connect internal damage to different retouched industries is intriguing, especially in Quina contexts.

Reviewer #2: The manuscript presents the study of bone shaft fragments interpreted as bone tools from the Quina Mousterian US22 of the Chez-Pinaud site (Jonzac, France). I think the issue is relevant, as the use of bone tools in Pre-Upper Palaeolithic/Later Stone Age (LSA) periods has been a controversial issue ever since Dart (1957) proposed the existence of the Osteodontokeratic industry. Studies of bone tools have become important for understanding the technological and cultural development of human groups in the past. Thus, bone technology should be included as one piece of the “modern human behavioural" repertoire. I consider that the study deserves publication due to the topic it deals with, however some critical points should be addressed before its publication.

Bone has been used for many purposes since the beginning of the Pleistocene. Bones used for shaping stone tools are prevalent in late Lower Palaeolithic Europe and in the Levant as early as MIS 13. So, I don't think the question "Did Neanderthal produce a bone industry?" with which the authors begin the Abstract is appropriate. There are several pieces of evidence that show the use of bones not only by Neanderthals, but also by earlier hominids, and for this reason I advise the authors to delete this very strict sentence and relativize the exceptionality that Neanderthals used bones as tools. I think it is important that the authors reformulate the beginning of the Introduction making a distinction in the use of bones during the Pleistocene from a technological point of view by differentiating between 1) intentionally polished bones, 2) bones knapped by direct percussion (retouched edges or flaked), and 3) unmodified bones used for a particular purpose. I think the polishing involves an important step in the making and handling techniques of bone artefacts and maybe it could be highlighted.

It seems to me that the study is clear on a methodological and comparative level. However, I have to say that some bone morphologies shown in the manuscript are similar to those that resulted from my experiments in breaking bones to access the marrow with fresh bone. In my experience, bone morphologies can be highly variable depending on variables such as the size, type and morphology of the hammer-stone, the state of the bone at the time of breakage or the areas that are not clean of periosteum [an irregular distribution of the periosteum could produce differences in the propagation of the impact force on the bone, generating scalariform breakage planes, jagged fracture surfaces and sometimes beveled fractures (like e.g., fig.7,8)]. With this I do not mean that the interpretation of the authors is wrong, but rather that sometimes the variables play a critical role in the resulting morphologies, and especially in the cases in which the periosteum is only removed from some specific areas. In fact, the authors point out the importance of the variables on p.14. For this reason, I encourage the authors to continue along this working-line, expanding the experimental series and variables in a near future.

I would like the experiments to be explained in detail in the supplementary material (S5Text), especially those focused-on plant working (e.g., herbaceous harvesting), skin working (e.g., proto-tanning) and soil working (e.g., digging up roots), and see the specific bone damage associated with these activities (types of marks, affected areas, etc.) and the distinctive features depending on the development of one activity or another. I would like to know variables used (e.g., trained and non-trained individulas, duration of the process, etc.). I think the details in the supplementary material would greatly enrich the scientific story line.

In the S4Text the authors state: “The fractures considered as anthropogenic are those occurring on "fresh" bone, i.e., occurring shortly after the animal's death”. This statement should be clarified or slightly modified because human groups can also use and work dry or semi-dry bone.

As a curiosity, the presence of the fox arouses my interest. Do the fox bones have any marks that allow their presence to be related to some human or carnivore activity? Is it a natural intrusion? What anatomical parts are represented?

6. PLOS authors have the option to publish the peer review history of their article (what does this mean?). If published, this will include your full peer review and any attached files.

Reviewer #1: **Yes: **Eva Francesca Martellotta

Reviewer #2: No

---

## [Author Response · Author response to Decision Letter 0]

28 Feb 2023

Response to Reviewers

(Auth) We are very grateful to the Academic Editor and Reviewers for their interest in our study and for taking the time to read the manuscript and suggest corrections for improvement. Their recommendations were helpful and we followed them. Where we have not, we have explained why. We hope that the many changes to the main text make our objectives and results clearer.

Academic editor comments:

(AE) Both reviewers consider this work worthy of publication although in the meantime they request that you tackle some issues in the structure of the article. These particularly affect the first part of the manuscript, where unnecessary data, unclear sentences and statements hamper the fluency of reading. Moreover, you are invited to clearly state which gaps currently concern the study of bone technologies in the Middle Palaeolithic and to avoid incertitudes about the criteria you used to isolate the bone fragments from the faunal assemblage, to previously identify them as tools and using them as materials for the whole study, given the lack of common criteria previously applied by different scholars. Reviewers also requested to better explain the experimental trials and re-arrange the discussion section. Therefore, we invite you to submit a revised version of the manuscript that addresses the points raised during the review process.

(Auth) Our proposals to address these issues are outlined following the reviewers’ comments.

Reviewer #1: 

Introduction

(R1) This is the most problematic part of the manuscript. Although the authors give an excellent framework of the gaps currently affecting the study of Middle Palaeolithic (MP) bone technologies, they do not state clearly enough what is their proposal to fill these gaps. Is the aim of this research to address the lack of ‘shared criteria’ in the study of MP bone industries by proposing a new analytical methodology (micro-CT)? 

(Auth) We do not propose a new methodology, but apply a classical techno-functional analysis as defined by S.A. Semenov (1964) and developed after him by many authors. However, as this type of analysis has rarely been applied to Neanderthal bone assemblages, it is likely to bring new results. In order to clarify this point, we modify the text lines 90-91 and lines 107-108 and complete the methodology section with elements from the supplementary data. The use of Micro-CT for the study of bone tools can be considered a new methodology, but we do not wish to highlight it in this article, as its contribution is limited, mainly to document the objects and allow more complete comparisons than with photographs (as done by Soressi et al. 2013). A dedicated article explaining the means and objectives of the "internal traceology” is in progress.

(R1) More specifically, is it to give guidelines to other researchers to identify (in the words of the authors) a ‘genuine bone industry’? 

(Auth) The main objective is simply to enrich the archaeological and experimental referentials, which are still few for the Middle Paleolithic. We modify line 99 the text.

(R1) Or is it to replicate the methods previously proposed by the authors (Chagyrskaya) and to apply those methods to a Neanderthal context in Western Europe, since Chagyrskaya and Chez-Pinaud are both Quina sites? 

(Auth) Chagyrskaya is an Eastern Micoquian site. We add a paragraph in the introduction that gives more details about the site and the bone industry study that was conducted there. The first step is to see if the bone tool production was a regular practice, a behavior common to all Neanderthals. We chose a site with a distinct lithic tradition (Quina) to ensure that the Chagyrskaya bone industry is not just a regional specificity of the Eastern Micoquians, as specified in the introduction line 106-107.

(R1) Some structure and rephrasing are necessary. I suggest, among other things, (1) giving a clear, technical definition of what the authors mean by ‘bone tool’ and ‘bone industry’, 

(Auth) We gave our definition of what we mean by industry in the second paragraph of the introduction, lines 62-63: “their number, diversity and recurrence lead to their consideration as an industry”. In order to make this clearer, we have added a clarification directly after. For the bone tool, please, see our response below.

(R1) (2) giving a summary of methods and results applied in the study of the Chagyrskaya bone industry, and state why we should follow a similar approach for Chez-Pinaud, and

(Auth) As the results of the analysis of the Chagyrskaya bone industry are already published and available online, we think it would be redundant to repeat them here, and we do not want to confuse the reader about the topic addressed. This is an article about the material from the Chez-Pinaud site and not a systematic comparison of the material from Chez-Pinaud with that from Chagyrskaya. The data on Chagyrskaya are mobilized only in the discussion. We have added information about the Chagyrskaya bone industry in paragraph 2 of the introduction.

(R1) (3) use a more dubitative tone throughout the Introduction, to communicate that previous works do exist and are valid, but need updating. 

(Auth) In the introduction, we do not question the validity of previous works. We simply propose hypotheses that could explain why some of the tools may have been disregarded. In most of the other works, the number of reported pieces is low apart from the bone retouchers. We simply want to draw attention to the other categories of tools that, we believe, have not been noticed. We have modified the text of lines 90-94 accordingly to clarify this point.

(R1) The authors have done excellent work with the Supplementary Information, and most of what is missing from the manuscript can be taken from there.

(Auth) We follow this recommendation and integrate much of the supplementary data in the manuscript (Methodology section).

(R1) Following, are some specific comments:

• Line 66: the sentence is unclear. Do you mean, it is the only bone industry surely made by Neanderthals? 

(Auth) We modify the sentence to clarify our point. Instead of “To date, the Chagyrskaya bone tools are the only example of a Neanderthal bone industry, at least, the only one whose authorship is not debated”, we write “To date, the Chagyrskaya bone tools are the only example of a Neanderthal bone industry (production of a set of tools), at least, the authorship of Anatomical Modern Human cannot be considered”.

(R1) This is the part where I believe a clear definition of ‘bone tools’ would be appropriate. In lines 65-81 there is a list of previously identified bone tools ‘categories’, but it results as a mix of retouchers, lissoirs, scrapers, etc without ever addressing the general definition of ‘bone tool’. 

(Auth) We are not sure we understand this comment. Do you think that retouchers, smoothers, scrapers etc. are not bone tools? The word “tool” is used in its general acceptation. “A tool is any instrument or simple piece of equipment that you hold in your hands and use to do a particular kind of work” (Collins Online Dictionary). More simply, it can be defined as an intermediary for action. In this case, it is made of bone. This is a technical definition. Nevertheless, the recurrence of a particular morphology adapted to a functional requirement (by selection or shaping) leads to the formation of a type of tool. 

(R1) We all agree that Neanderthals can make a ‘proper’ bone industry -- but what is a ‘proper’ bone industry in fact? Because it is such a big question, citing previous research addressing this issue might not be enough in this case.

(Auth) Is your comment referring to lines 99-100? The issue is not about Neanderthal's ability to make a bone industry (for us, it is neither a question of cognition nor of modernity), but about the fact that he actually did it. We did not write "proper" but "genuine". The argument we develop is that 1) with the exception of Chagyrskaya, the diversity of Mousterian bone tools has not yet been reported on a single site (whereas at Jonzac we are in an episode of occupation of short duration and over a few square meters) 2) this low visibility is the consequence of a shaping by percussion.

(R1) Line 78: ‘most of them recently investigated’ might be more appropriate.

(Auth) Done

(R1) Line 91: I would rather say that a proper study of MP bone tools (not bone industry) is anecdotal. It might be true that bone industries are only sporadically present in the MP -- we actually don’t know it because we never created a methodology to investigate bone tools! I believe the key to this sentence should be to critique the previous approaches to the study of bone tools, in fact anecdotal and most of the time borrowed from lithic studies.

(Auth) We have modified the sentence to better explicit the link between the two assertions. 

(R1) Line 92: ‘mode of manufacturing’ sounds confusing. Maybe ‘manufacturing techniques’? There is also a grammatical discrepancy (plural ‘their’ / singular ‘bone’).

(Auth) Done for "techniques". "Their" refers to the tools (previous sentence), while "bone" refers to the material.

(R1) Line 94-96: give some literature references to other studies highlighting the issue of the lack of shared identification criteria (some are cited elsewhere in the Introduction).

(Auth) We have modified the sentence.

(R1) Line 98: briefly summarise the results from your previous study. Moreover, ‘evidence in the literature’ might be more appropriate than ‘bibliographic data’.

(Auth) Done. We have added a short summary of our results in the second paragraph of the introduction.

(R1) Line 99: ‘genuine’ is somehow a foggy term, I suggest rephrasing it.

(Auth) Done 

(R1) Line 100: ‘primary use of percussion’ is also a little foggy. What do you mean by ‘primary’?

(Auth) We have replaced “primary” by “predominant”.

(R1) Line 107-108: like in my previous comment (line 98) please openly state what the ‘Chagyrskaya analytical framework’ is.

(Auth) We have replaced “Chagyrskaya analytical framework” by “a techno-functional analysis”.

(R1) Line 108: something like ‘a site characterised by Neanderthal occupation’ would sound better to the reader.

(Auth) We replace “Neanderthal site” by “Neanderthal settlement”.

(R1) Line 111: here ‘bone tool’ should be replaced with ‘bone industry’.

(Auth) Done

(R1) Line 116-118: it is unclear to me why we should compare Chagyrskaya and Chez-Pinaud. If you want to apply the same methods from Chagyrskaya, why did you choose Chez-Pinaud and not any other site with Neanderthal occupation and previously identify bone retouchers and bone-bed?

(Auth) Theoretically, the Chagyrskaya site could be compared to any Western European sites since the first objective is to identify a Neanderthal bone industry in the western part of Eurasia (to ensure that the production of a bone industry is a behavior that can be observed on a large geographical scale), and in another Middle Paleolithic techno-cultural facies (to ensure that the production of a bone industry is not only a specificity of the Siberian Eastern Micoquian Neanderthals) as specified in lines 105-108 and 116-118. In practice, the Chez-Pinaud site provides one of the best models:

- continuous excavations providing high quality of contextual data

- no possible contamination of Upper Paleolithic levels

- a large quantity of bone (bone-bed) that allows to observe recurrences. By the way, there are not many Middle Paleolithic sites with beds bone 

- few disturbances (remains in anatomical connection)

- numerous retouchers identified during previous excavations. In our opinion, this is a strong indication of the presence of a bone industry

- in-depth studies of the faunal remains (5 zooarcheologists involved)

Methods

(R1) In general, I would appreciate more depth in the methods, since the main aim of the authors is supposed to be to encourage the creation of a shared methodology for the study of bone tools. Particularly, I am confused about the criteria the authors used to isolate the 103 bone fragments from the faunal assemblage by giving them the definition of ‘tools’ and using them as materials for the whole study. As we understand from the Introduction, we do not have any shared criteria to do so at the moment, and the authors need to share the criteria they used. Most of the requested information is in the SI, but I believe stronger statements are needed here in the manuscript. The same goes for details on the experimentally produced bone tools. I do not think the whole experimental protocol belongs to the manuscript, but perhaps some justification of why you need the experimental bone tools within your analysis might fit well in the Methods section.

(Auth) As explained in previous responses, our aim is not to provide a new methodology or new criteria. We have modified the sentence on lines 94-96. Instead of “ the lack of shared identification criteria makes comparisons between sites difficult” we write “the diversity of published identification criteria and vocabulary make comparisons between sites difficult”. Our criteria are those of traceology (sensu S.A. Semenov). We have completed the methodology section.

(R1) Line 138-140: unclear sentence. Does it mean that during the last excavation lithics were recovered and subjected to preliminary studies which revealed they belong to Quina industries?

(Auth) We have changed the sentence. Instead of “are in agreement with previous studies” we write “are in line with the results of previous excavations”

(R1) Line 147-148: are there any other indicators of the cold, dry and open environment, apart from reindeer remains? If so, please cite them. Moreover, I am confused if these data refer to US 22 only, or a general description of the occupation of the site.

(Auth) We have added “of the US22”. We do not describe the site environment but only pointed out that the reindeer is a marker of a cold, dry, open environment. We have added two bibliographic references. 

(R1) Line 158-159: ‘bones anatomical connection’. Unclear. Do you mean ‘bones refitting’ or ‘bones which are anatomically connected (articulated) in situ’?

(Auth) We specify “in situ”

(R1) Line 182-183: it seems to me somehow circular logic here. If the aim of the study is to identify bone tools among faunal assemblage by using a specific methodology, how can we state, in the methods, that 103 bone tools were found? Maybe this sentence belongs to the Results section.

(Auth) We have removed this sentence (“Through this process, a total of 103 bone tools were found”) from the material and method section and have added the information at the beginning of the results section.

(R1) Line 202: what are the criteria used to select these 9 bone tools for the micro-CT? Did you choose one tool per ‘bone category’? Is it because of better preservation? Specific spatial distribution in the site? Please be more specific.

(Auth) We have added a sentence to explain our choice.

Results

(R1) I am confused by the description of bone tools from recent and old excavation campaigns. Do the results concern only recent findings (US 22), or do they also re-examine bone tools previously identified in older campaigns, but with the new ‘methodological framework’ proposed by the authors? Some clarification is needed.

(Auth) The first sentence of the material and method section clearly states: “ The bone tools studied are those from the first two excavation campaigns (2019-2020) conducted by W. Rendu, K. Kolobova and S. Shnaider. They come exclusively from US22, attributed to the Quina Mousterian”. 

(R1) If, in fact, they do re-analyse bone tools previously identified, the authors should take time in the discussion to assess the validity of their new proposed methodology against the ‘unshared criteria’ used until now in the study of MP bone technologies. If the results of the present work mainly focus on bone tools coming from recent excavation, then I suggest deleting any reference to tools collected in previous campaigns, to avoid confusion. Perhaps in the future, the authors might want to consider undertaking a comparative study with bone tools described in the past to determine how their method is an improvement on what people did before, and how we can build new, shared criteria for the study of bone tools based on the methods the authors propose.

(Auth) Again, we do not propose a new methodology. Archeozoologists and lithic specialists are able to recognize bone retouchers, sometimes retouched bone tools or smooth-ended bone tools, at least the most obvious specimens. The only difference, here, is that the bone assemblage from recent campaigns (2019-2020) is examined by a specialist in techno-functional analysis of bone tools, whereas the faunal assemblage from earlier campaigns was sorted by faunal specialists only. We believe it is relevant to mention the earlier works, as they contribute to the characterization of the level studied. 

(R1) Moreover, I found myself confused while reading the results about use-wear and manufacturing traces descriptions: it is not always clear if the authors are talking about traces observed on archaeological bones, rather than experimental bones, rather than results obtained in previous studies. Perhaps a more structured organisation of this section might be helpful, and comparisons within traces and tools might find a more suitable space in the Discussion.

(Auth) We have modified the first sentence of each paragraph of the results to specify more clearly (when it is not already done) if we speak about our archeological or experimental specimens. Concerning the “external modifications”, the descriptions are those of the archeological specimens. Concerning the “internal damage”, one paragraph is devoted to the experimental specimens while the next one is devoted to the archeological specimens, because we do not know a priori what kind of internal damage can occur in the bone tools, a description of experimental specimens is necessary first. The paragraphs on the smooth-ended tools are only about the archeological specimen, because so far, we do not have a close enough experimental comparison. 

(R1) Fig. 2: a more schematic organisation of the identified tool ‘types’ might help the reader to get a visual understanding of the differences between the ‘types’ themselves.

(Auth) It is not possible to strictly organize the plate according to the different types of bone tools since there are multiple-tools, as noted in lines 227-230. For example, the first bone tool is a beveled tool, with lateral retouches, also used as retouchers.

(R1) I question the use of the word ‘type’ to identify a tool showing specific features on which we can base the tool’s function (retouchers, smooth-ended tools, etc.). To me, the term ‘type’ carries too many reminiscences from the lithic typological approach (Bordes sensu), which several scholars consider nowadays an obsolete approach even for lithic technology. More importantly, it could be argued that the use of this terminology is inconsistent with what the authors stated in the Introduction, blaming the lack of proper methods for the study of bone tools on reckless borrowing from pre-existing lithic studies. The same goes for ‘tool category’. I suggest the authors might reconsider terminology here; for instance, in other parts of the manuscript, the authors use the term ‘morpho-functional category’, which I believe is the most appropriate definition the authors used so far.

(Auth) The word “type” is used in its general acceptation: “group of things that have particular features in common” (Collins Online Dictionary). It is a basic term used in archaeology (sensu D. Clarke, 1968). What is obsolete is to base a typology only on the shape of objects without distinguishing their state and position in the process of production and use (chaîne opératoire). Our types are based on technological criteria.

(R1) Line 255: a picture of these percussion marks and notches, even only in SI, could be appropriate. It could contribute to the debate regarding the attribution of percussion marks and notches to butchering activities rather than the use of bone blanks as retouchers (see: Delphine Vettese, Camille Daujeard. Le mythe du retouchoir en os : analyse d'un reste actuel / The myth of the bone retoucher: analysis of an actualistic bone referential. Colloque en hommage à Emilie Campmas "Sociétés humaines et environnements dans la zone circumméditerranéenne du Pléistocène au début de l'Holocène", Mar 2021, Toulouse, France).

(Auth) We are not sure we understand this comment. It is difficult to consider work that has not yet been published in a peer-reviewed journal. The data presented in the poster you refer to is not sufficient to discuss the proposal. Our identification of percussion marks and notches on the one hand and bone retouchers use-wear traces on the other, is based on abundant published data and our own corpus (about 80 long bone breaking and more than 100 bone retouchers), both of which we believe are substantial enough not confuse the two sets of marks. Under the conditions described by the poster, on a single example, an overlap at the margin of the two distinct populations of impact features cannot of course be ruled out, but so far this does not seem representative of the trend. 

(R1) Line 261-268: a comprehensive table of these metric data in SI might help future researchers to compare their bone industry with yours.

(Auth) Lines 261-268 refer to the faunal remains.

(R1) Line 280-282: tibias indeed are the easiest to recognise, but they are also the easiest bone to fracture and the bone that usually produces the greatest number of blanks upon breakage. Perhaps your observations are not biased…

(Auth) In our experiments, the other long bones seem no more difficult to break (except perhaps the radio-ulna when the ulna is fused with the radius). The number of fragments you get depends on the level of breakage you choose depending on the type of blanks you need. 

(R1) Line 283 (and 289, 292, 316, 381…): I would move all the comparisons with other Quina sites in Discussion.

(Auth) Line 283: We have removed the sentence.

Line 289, 292 and 316: The paragraphs form a coherent unit in which it would be difficult to separate the raw data from the comparison. Since this is a one-time information about retouchers, we prefer to keep it in the dedicated section, rather than including the raw data in the discussion, which refers to more general considerations.

Line 381 is not relevant to the comparison with other Quina sites.

(R1) Line 304-306: same -- I would move comparisons with retouched lithic tools in Discussion.

(Auth) Done

(R1) Line 331-343: this section might belong to the Methods.

(Auth) As this is the first time that such an analysis has been carried out, we consider these (our) proposals and observations to be part of the results.

(R1) Line 346: please give a definition of ‘simple and complex micro-cracks’ in the Methods.

(Auth) Do you mean a definition of “cracks” or “simple and complex”? For the latter, the explanation is given in the sentence « They are mainly simple, but can also be complex, with one or more secondary cracks from a main crack”. “Crack” is used is in general acceptation: “a very narrow gap between two things, or between two parts of a thing” (Collins Online Dictionary).

(R1) Line 364-375: explanatory parts of this section should be moved to the Methods, whereas actual data and results should be left here.

(Auth) Please, see response to the comment “line 331-334”

(R1) Line 390-391: do you have any experimental evidence for this fracture to be resulting from the tool’s use?

(Auth) These are well-known fractures on bone artifacts. We have added references.

(R1) Line 412-419: this should be in the Discussion.

(Auth) These lines introduce the description of two particular tools.

(R1) Can the authors spend a few words explaining how we can distinguish the manufacturing marks associated with ‘retouched tools’ from the percussion marks associated with the bone breaking during butchering activities? Retouched bone blanks are widely present in Quina sites, and a reliable traceological identification protocol is needed.

(Auth) We have added a paragraph in the Methods section about the experimental retouch and a few words explaining how we distinguish them from the percussion marks from butchering activities line 503. See also Text S3. For the active retouched edges, the retouched area is associated with use-wear traces (described in each case). 

(R1) Line 533: can you elaborate on this hypothesis?

(Auth) We have corrected the sentence (translation mistake).

(R1) Line 549: please reference previous studies or the authors’ experimental evidence for this use-wear in association with cutting soft materials.

(Auth) It is basic knowledge for use-wear analyst that the degree of wear of an active edge is directly correlated to the limit of its effectiveness on the material being worked, while the orientation of the linear marks is generally indicative of the kinematics, as are edge removals. There is nothing specific to bone material here. Thus, the dulling of an edge in longitudinal motion cannot result from processing a hard or semi-hard organic material. On our archaeological sample, it is also too discernible by comparison with our experimental references to be the result of a meat cutting, but fresh skin and fascia may be good candidates. Because we cannot rule out the presence of soft plant material (but the coverage would likely be different), we do not specify whether it is animal or vegetal material. 

(R1) Line 601-602: ‘including the use of the retouched edge’. I would like to see an experimental test, followed by micro-CT, of the retouched edge after the use proposed by the authors. The discussion about internal damage due to use-wear and use of the retouched edge is fascinating and deserves to be verified.

(Auth) The purpose of this paper is to test the possibility of using Micro-CT analysis to detect internal damage in archaeological bone tools. The precise characterization of this damage will be the next step of our work.

(R1) Line 620: ‘must’ might be a little strong…

(Auth) We have modified the sentence.

(R1) Line 649: this claim is a little too blunt and needs more justification. What use? Which bone? Which material is being worked?

(Auth) All information is given in the sentence: “Experimentally, the sharpness of a bone edge decreases quite rapidly during its use, but its loss of matter is minimal unless the material being worked is abrasive”. This is a general (for cortical bone tissue) and basic observation.

Discussion

(R1) After working on the previous sections, I believe there will be several topics the authors will want to address in the Discussions. At the moment, this section is somehow meagre and disorganised. I suggest giving a proper structure to this section through thematic headings (e.g., comparisons with other Quina sites, the potential of the analysis of internal damage, multifunctionality/re-use of bone tools in Quina culture compared to the lithic industries, suggestions for the creation of ‘shared criteria’ to study bone tools…).

(Auth) We have added subheadings to the discussion section to make its structure clearer. 

(R1) Line 685: why is it important to state that the number of bone tools is equal to the number of lithic tools?

(Auth) It emphasizes the quantitative importance of the bone tools. It is a way of suggesting that they should be analyzed in the same way as lithic pieces, i.e., as an industry.

(R1) Line 753-754: I would like to hear more details on this. Why not? What would be a better investment instead?

(Auth) We have modified the sentence and added references.

(R1) Line 763-764: what is your hypothesis on this, based on your results, observations and comparisons with other sites? You somehow address it in lines 784-789, but a little of a ‘dreaming’ speculation here couldn’t hurt.

(Auth) The complementarity mentioned in lines 784-789 applies more to tools with the beveled and smooth ends than to the lateral cutting edges. As this time, we have no tangible hypothesis to put forward. 

(R1) Line 766-773: very interesting!

(Auth) Thank you

(R1) Line 778: ‘pre-Upper Palaeolithic’ might be misleading. Please rephrase.

(Auth) Sorry, we do not understand this comment. “pre” is using as an equivalent of “before” or “priori to”.

(R1) Line 801: typo. Replace ‘narrow’ with ‘marrow’.

(Auth) Done

(R1) Line 1255: correct spelling is ‘Martellotta’ :)

(Auth) Done. Sorry.

(R1) In conclusion, I believe this is much-needed work because we require a shared methodology in the study of MP ‘non-official’ bone tools, as well as a reliable methodology to assess the multifunctionality of these tools. Throughout the whole paper, I very much appreciated the focus on traceological descriptions and mechanical properties of osseous materials. The application of the micro-CT to traceological questions is a fascinating subject, and its brilliance could and should be more emphasised in the paper. The authors’ proposal to connect internal damage to different retouched industries is intriguing, especially in Quina contexts.

Reviewer #2: 

(R2) The manuscript presents the study of bone shaft fragments interpreted as bone tools from the Quina Mousterian US22 of the Chez-Pinaud site (Jonzac, France). I think the issue is relevant, as the use of bone tools in Pre-Upper Palaeolithic/Later Stone Age (LSA) periods has been a controversial issue ever since Dart (1957) proposed the existence of the Osteodontokeratic industry. Studies of bone tools have become important for understanding the technological and cultural development of human groups in the past. Thus, bone technology should be included as one piece of the “modern human behavioural" repertoire. I consider that the study deserves publication due to the topic it deals with, however some critical points should be addressed before its publication. Bone has been used for many purposes since the beginning of the Pleistocene. Bones used for shaping stone tools are prevalent in late Lower Palaeolithic Europe and in the Levant as early as MIS 13. So, I don't think the question "Did Neanderthal produce a bone industry?" with which the authors begin the Abstract is appropriate. There are several pieces of evidence that show the use of bones not only by Neanderthals, but also by earlier hominids, and for this reason I advise the authors to delete this very strict sentence and relativize the exceptionality that Neanderthals used bones as tools. 

(Auth) “Did Neanderthal produce a bone industry?” is precisely the question at the heart of our work. We know that there is evidence for the use of bone for tools manufacturing by Neanderthals, as discussed in the third paragraph of the Introduction. However, there is a significant difference between the regular manufacture of bone tools for a variety of purposes (i.e., a bone industry) and the opportunistic use of bone for an expedient task, as considered by most previous studies due to the very small number of identified Mousterian bone tools other than retouchers. We agree that there is nothing exceptional about finding bone tools in Neanderthal assemblages, as we believe that this is a common behavior, part of the Neanderthal technical repertoire. Our aim is to draw attention to the pieces that we believe have been disregarded. They change the perspective. If we want to understand the specificity of Neanderthal bone productions, either technologically or in terms of activities, we need to make sure that we have all the components of that production, which we believe is not yet the case at most of the sites where bone tools (retouchers or otherwise) have been reported to date.

(R2) I think it is important that the authors reformulate the beginning of the Introduction making a distinction in the use of bones during the Pleistocene from a technological point of view by differentiating between 1) intentionally polished bones, 2) bones knapped by direct percussion (retouched edges or flaked), and 3) unmodified bones used for a particular purpose. I think the polishing involves an important step in the making and handling techniques of bone artefacts and maybe it could be highlighted.

(Auth)This article deals exclusively with the Neanderthal case. It does not address the evolution of bone tools during the Pleistocene. In the context of our study, the distinction between “intentionally polished bones”, “bones knapped by direct percussion”, and “unmodified bones used for a particular purpose”, does not seem relevant since these different categories are present in the same industry (here, there are strictly contemporary). They should be considered and analyzed as a technically coherent whole. We do not believe that the use of “polishing” (abrasion) technique is in itself a fundamental step. This is an issue we have addressed in a previous paper (Baumann et al. 2022, Not so unusual Neanderthal bone tools: new examples from Abri Lartet, France, AASC).

(R2) It seems to me that the study is clear on a methodological and comparative level. However, I have to say that some bone morphologies shown in the manuscript are similar to those that resulted from my experiments in breaking bones to access the marrow with fresh bone. In my experience, bone morphologies can be highly variable depending on variables such as the size, type and morphology of the hammer-stone, the state of the bone at the time of breakage or the areas that are not clean of periosteum [an irregular distribution of the periosteum could produce differences in the propagation of the impact force on the bone, generating scalariform breakage planes, jagged fracture surfaces and sometimes beveled fractures (like e.g., fig.7,8)]. With this I do not mean that the interpretation of the authors is wrong, but rather that sometimes the variables play a critical role in the resulting morphologies, and especially in the cases in which the periosteum is only removed from some specific areas. In fact, the authors point out the importance of the variables on p.14. For this reason, I encourage the authors to continue along this working-line, expanding the experimental series and variables in a near future. 

(Auth) Here, the identification of the bone tools is not based on the morphology of the bone fragments but on the combination of use-wear traces and technological features. In the case of the beveled tools, the morphology of the active end is of course part of the identification criteria. However, it is not sufficient to validate the status of the tool. For each beveled tool we have described what we consider to be anthropogenic modifications based on published data (use-wear analysis and bone technology), our own experimental frame of reference and the expertise of 3 techno-traceologists and 2 zooarcheologists. We have enriched the “Methods” section with data that were previously included in the supplementary data. The "Experimental frame of reference" section provides an overview of the experiments conducted since 2019 when we began studying the bone assemblage from the Chez-Pinaud site, but our referential also includes previous experiments with bones in different states of freshness, different hammers, different way of fracturing etc. We are well aware of the morphometric variability, and we address the issue, for example, when we discuss the question of fracture planes angles.

(R2) I would like the experiments to be explained in detail in the supplementary material (S5Text), especially those focused-on plant working (e.g., herbaceous harvesting), skin working (e.g., proto-tanning) and soil working (e.g., digging up roots), and see the specific bone damage associated with these activities (types of marks, affected areas, etc.) and the distinctive features depending on the development of one activity or another. I would like to know variables used (e.g., trained and non-trained individulas, duration of the process, etc.). I think the details in the supplementary material would greatly enrich the scientific story line.

(Auth) The publication of the whole experimental frame of reference, even preliminary, would probably fill more than a hundred pages of supplementary data. This long work of systematization, with macroscopic photographs of the original pieces, microscopic photographs of epoxy replicas, 3D modeling of volumes and internal deformations is intended for another type of media. It constitutes a subject in itself while the supplementary materials are a kind of unreferenced literature. 

(R2) In the S4Text the authors state: “The fractures considered as anthropogenic are those occurring on "fresh" bone, i.e., occurring shortly after the animal's death”. This statement should be clarified or slightly modified because human groups can also use and work dry or semi-dry bone.

(Auth)This statement is clarified in the following sentences: “However, the transition from fresh to dry bone is a gradual and non-homogeneous phenomenon. Therefore, the proposed criteria do not allow for the recognition of an anthropogenic fracture on drying bone or a post-depositional fracture on fresh bone. In the archaeological material, the fracture surfaces often have mixed features, which implies reasoning in terms of trends according to the corpus studied and taking into account all available discriminating criteria (hammers impacts, general morphology, location, variability within the corpus, etc.).”

(R2) As a curiosity, the presence of the fox arouses my interest. Do the fox bones have any marks that allow their presence to be related to some human or carnivore activity? Is it a natural intrusion? What anatomical parts are represented?

(Auth) The fox remains found during the 2019-2020 campaigns do not show evidence, but a distal end of a fox tibia found in 2007 (Jaubert/Hublin excavations) show evidence of cut marks (see Niven et al. 2012).

---

## [Decision Letter · Decision Letter 1]

14 Mar 2023

PONE-D-22-30527R1On the Quina side: A Neanderthal bone industry at Chez-Pinaud site, FrancePLOS ONE

Dear Dr. Baumann,

Thank you for submitting your manuscript to PLOS ONE. After careful consideration, we feel that it has merit but does not fully meet PLOS ONE’s publication criteria as it currently stands. Therefore, we invite you to submit a revised version of the manuscript that addresses the points raised during the review process.

We look forward to receiving your revised manuscript.

Kind regards,

Marco Peresani

Academic Editor

PLOS ONE

Journal Requirements:

Reviewers' comments:

Reviewer's Responses to Questions

**Comments to the Author**

1. If the authors have adequately addressed your comments raised in a previous round of review and you feel that this manuscript is now acceptable for publication, you may indicate that here to bypass the “Comments to the Author” section, enter your conflict of interest statement in the “Confidential to Editor” section, and submit your "Accept" recommendation.

Reviewer #1: All comments have been addressed

2. Is the manuscript technically sound, and do the data support the conclusions?

Reviewer #1: Yes

3. Has the statistical analysis been performed appropriately and rigorously? 

Reviewer #1: Yes

4. Have the authors made all data underlying the findings in their manuscript fully available?

Reviewer #1: Yes

5. Is the manuscript presented in an intelligible fashion and written in standard English?

Reviewer #1: Yes

6. Review Comments to the Author

Reviewer #1: I am grateful to the authors for their responses to my concerns on the manuscript ‘On the Quina side: A Neanderthal bone industry at Chez-Pinaud site, France’.

Their responses clarified the main aim of the study, which was also clarified within the text. Although the first version of the manuscript was misleadingly drawing attention on the applied methodology, this new version has been toned down and it is more consistent with the actual aims of the authors. The same can be said for the role of the Chagyrskaya site in the context of present research, made much clearer by the authors in the revised manuscript.

Moreover, the manuscript is now much more structured, and the numerous details added by the authors fit well in the new structure of the article. I appreciated the addition of many details concerning the experimental protocol, inasmuch it makes the manuscript now consistent with the publication requirements of PLOS ONE.

In conclusion, I recommend the article for publication, however the text could beneficiate from a reread in order to address typos and rephrase some foggy sentences. I suggest here the ones I noted while reading.

Lines 67-68: Grammar here could be improved. Do you mean ‘To date, the Chagyrskaya bone tools provide the only example of a Neanderthal bone 68 industry *for which* the authorship of AMH cannot be considered’?

Line 67-69: I am confused by this statement. If Chagyrskaya is the *only* example of a Neanderthal bone industry, how can you state just after that it is *not an isolated* case?

Lines 80-81: This sentence is unclear. What does ‘others’ refer to? Other discoveries? Other industries? Please rephrase.

Lines 283-284: rather than ‘close’ striking angle I would say ‘acute’.

Line 383: replace ‘Bos/bison’ with ‘Bos/Bison’ (capitalised words and italic).

Line 415: ‘The US22’. Remove ‘The’.

Line 467: Adjust the grammar here. Change the sentence to either ‘the mode of use of *a* bone retoucher […]’ or ‘the mode of use of bone *retouchers* can also be revealed by *their* internal damage.

Line 468: replace ‘reciprocal effect of one on the other’ with ‘reciprocal effect on one another’

Line 564: replace ‘in the main axis’ with ‘along its main axis’.

Line 596: replace 2 and 4 with ‘two’ and ‘four’.

Line 710: replace ‘done by percussion’ with ‘produced by, resulting from, achieved through…’ percussion.

Line 853: correct spelling is ‘De Nadale’.

7. PLOS authors have the option to publish the peer review history of their article (what does this mean?). If published, this will include your full peer review and any attached files.

Reviewer #1: **Yes: **Eva Francesca Martellotta

---

## [Author Response · Author response to Decision Letter 1]

20 Mar 2023

Reviewer 1

Lines 67-68: Grammar here could be improved. Do you mean ‘To date, the Chagyrskaya bone tools provide the only example of a Neanderthal bone 68 industry *for which* the authorship of AMH cannot be considered’?

Done

Line 67-69: I am confused by this statement. If Chagyrskaya is the *only* example of a Neanderthal bone industry, how can you state just after that it is *not an isolated* case?

“The bone tools from Chagyrskaya are not an isolated case” have been replaced by "Bone tools have already been reported in Neanderthal sites but most of the time as isolated finds.".

Lines 80-81: This sentence is unclear. What does ‘others’ refer to? Other discoveries? Other industries? Please rephrase.

The sentence has been replaced by " These discoveries, most of them recently investigated, allow us to reconsider earlier ones which were potentially too readily dismissed"

Lines 283-284: rather than ‘close’ striking angle I would say ‘acute’.

Done 

Line 383: replace ‘Bos/bison’ with ‘Bos/Bison’ (capitalised words and italic).

We capitalized and italicized Bos (taxa) but not bison as it is the common vernacular name.

Line 415: ‘The US22’. Remove ‘The’.

Done

Line 467: Adjust the grammar here. Change the sentence to either ‘the mode of use of *a* bone retoucher […]’ or ‘the mode of use of bone *retouchers* can also be revealed by *their* internal damage.

Done

Line 468: replace ‘reciprocal effect of one on the other’ with ‘reciprocal effect on one another’

Done

Line 564: replace ‘in the main axis’ with ‘along its main axis’.

Done

Line 596: replace 2 and 4 with ‘two’ and ‘four’.

Done

Line 710: replace ‘done by percussion’ with ‘produced by, resulting from, achieved through…’ percussion.

Done

Line 853: correct spelling is ‘De Nadale’.

Done

---

## [Editor Report · Decision Letter 2]

23 Mar 2023

On the Quina side: A Neanderthal bone industry at Chez-Pinaud site, France

PONE-D-22-30527R2

Dear Dr. Baumann,

We’re pleased to inform you that your manuscript has been judged scientifically suitable for publication and will be formally accepted for publication once it meets all outstanding technical requirements. After careful reading and checking of reference list, figures, tables and materials of SM section, I noted very few flaws which have been highlighted in the annotated pdf attached to this notification. Required integrations are very few and I recommend to carry out them during proof cheking.   

Kind regards,

Marco Peresani

Academic Editor

PLOS ONE

---

## [Editor Report · Acceptance letter]

4 Apr 2023

PONE-D-22-30527R2 

On the Quina side: A Neanderthal bone industry at Chez-Pinaud site, France 

Dear Dr. Baumann:

I'm pleased to inform you that your manuscript has been deemed suitable for publication in PLOS ONE. Congratulations! Your manuscript is now with our production department. 

Kind regards, 

on behalf of

Dr. Marco Peresani 

Academic Editor

PLOS ONE